# CLASS-RELATIONAL LABEL SMOOTHING FOR LIFELONG VISUAL PLACE RECOGNITION

## ABSTRACT

Visual Place Recognition (VPR) is a task of estimating the location of a query image, predominantly executed through image retrieval using learned global descriptors from a reference database of geo-tagged images. While recent approaches have aimed to improve the scalability of VPR training by leveraging classification loss as a proxy task, this leads to a task gap between classification and retrieval – classification discretizes the feature space into distinct class regions, often overlooking visual differences between classes. This gap makes VPR systems particularly vulnerable to extreme visual changes such as lifelong variations. To remedy these problems, we propose a novel Class-Relational Label Smoothing (CRLS) that transforms one-hot labels into soft labels by considering visual information of inter-class relations. We further enhance this method by dynamically adjusting the influence of CRLS based on the stability of class weights, which is quantified by their magnitudes. Importantly, our findings suggest that the magnitude of class weights serves as an indicator of class stability, which is also supported by derivative analysis. We demonstrate that our method outperforms state-of-the-art methods on the most extensive 17 benchmarks, effectively bridging the task gap between classification and retrieval in visual place recognition. Code and trained weights will be made publicly available.

## 1 INTRODUCTION

Visual Place Recognition (VPR) is an important task in various applications, such as robotics (Stumm et al., 2013), autonomous driving (Bresson et al., 2017) and navigation (Mirowski et al., 2018), where its goal is to identify a location based on visual data. Typically, VPR is approached as an image retrieval problem (Wang et al., 2022; Zhu et al., 2023; Shen et al., 2023; Leyva-Vallina et al., 2023; Hausler et al., 2021; Arandjelovic et al., 2016; Warburg et al., 2020; Torii et al., 2013a; Thoma et al., 2020; Keetha et al., 2023; Lu et al.; 2024; Izquierdo & Civera, 2024), employing nearest neighbor search based on the similarity of descriptors between a query image and gallery images. This process enables the identification of the gallery image most similar to the query image, and subsequently, localization is achieved using the geo-reference of the identified image.

Real-world applications of VPR face numerous challenges due to significant appearance changes in various environments. These changes include well-known seasonal variations (Naseer et al., 2018; Sünderhauf et al., 2013), weather conditions (Ros et al., 2016; Berton et al., 2021), illumination changes (day/night) (Sattler et al., 2012; Maddern et al., 2017) and viewpoint changes (Carlevaris-Bianco et al., 2016; Berton et al., 2023). Subsequently, several benchmark works (Warburg et al., 2020; Ali-bey et al., 2022; Berton et al., 2022) have introduced lifelong datasets that include images collected over a long temporal span. We find that changes over extended periods, such as building modifications and remodeling, present extreme challenges for VPR, which we refer to as *lifelong variations*. For instance, Fig. 1a shows how two buildings evolve over time, with one highlighted in green and the other in orange. These buildings, located in San Francisco, undergo significant changes due to frequent urban modifications, often requiring continuous updates to the VPR models. To achieve effective retrieval for VPR, the trained model should continuously capture visual differences, leading to a continuous representation space that can deal with lifelong variations.

To learn such representation spaces, existing VPR methods (Arandjelovic et al., 2016; Berton et al., 2021; Hausler et al., 2021; Peng et al., 2021; Zhu et al., 2023; Leyva-Vallina et al., 2023; Wang

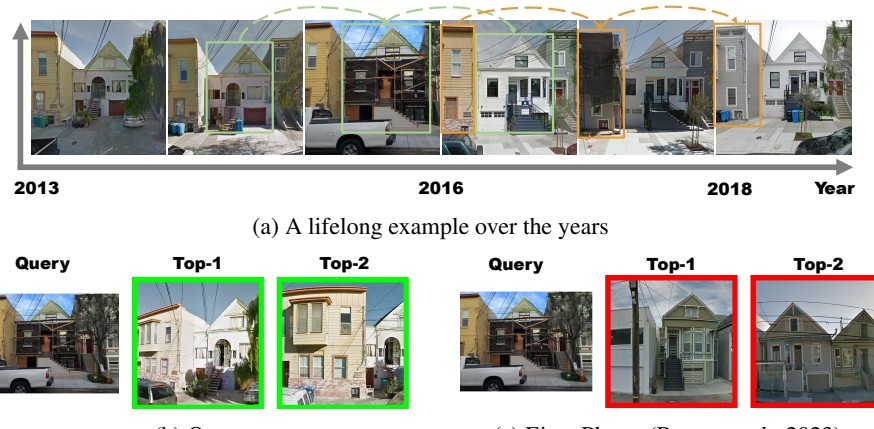

(a) A lifelong example over the years

(b) Ours                    (c) EigenPlaces (Berton et al., 2023)

Figure 1: **An example of lifelong visual place recognition** from SF-XL test v1 (Berton et al., 2022). (a) illustrates the evolution of buildings over time, using images of the same location from Google StreetView and Flickr, linked by provided GPS data. Lifelong scenarios often show significant changes due to frequent urban modifications. (b) and (c) compare retrieval results of our method and a state-of-the-art method, respectively. Positive images are highlighted in green, while negative images are highlighted in red.

et al., 2022) predominantly utilize metric learning losses such as contrastive or triplet loss, which rely heavily on mining negative examples throughout the training database (Arandjelovic et al., 2018). This mining process is notably resource-intensive and becomes prohibitively expensive as the size of the dataset increases. To address this scalability issue, more recent methods (Berton et al., 2022; 2023) have adopted a classification loss (*i.e.*, CosFace (Wang et al., 2018)) as a proxy task, thereby streamlining training by eliminating the need for the exhaustive negative mining. However, while this improves scalability, it introduces a task gap between classification and retrieval tasks. Specifically, classification tends to discretize the representation space into distinct class regions, leading to overconfident predictions (Guo et al., 2017). This contrasts with retrieval tasks, which require a continuous representation of subtle visual differences (Leyva-Vallina et al., 2021; Kim et al., 2021; 2022) to capture gradual and continuous changes in environments. A naïve approach to mitigate the issue can be label smoothing (Szegedy et al., 2016), which prevents overconfidence by bringing all classes closer together in the representation space (Müller et al., 2019). While label smoothing can partially relax the discretization of representation space, it assigns probabilities uniformly across all classes, disregarding the visual differences between them, making it less effective for retrieval tasks.

In this paper, we propose Class-Relational Label Smoothing (CRLS) for lifelong VPR to better address the task gap between classification and retrieval by incorporating visual similarities between classes into the label smoothing process, as illustrated in Fig. 2. Unlike conventional label smoothing, which treats all classes uniformly, we adjust the label distribution based on inter-class relationships, resulting in a continuous representation space where visually similar classes are closer, better reflecting subtle visual differences. However, the impact of CRLS may be influenced by fluctuations in class weights. To address this issue, we further introduce Class Stability Weighting (CSW), which dynamically adjusts the impact of CRLS based on the stability of class weights. Specifically, we empirically observe that class weight magnitudes reflect the classification difficulty of each class, as further supported by derivative analysis. Consequently, our method enables the model to learn a more robust and continuous representation that captures gradual visual differences over time, making it particularly effective in handling lifelong variations. Extensive experiments on 17 benchmarks demonstrate that our approach effectively bridges the task gap between classification and retrieval, achieving superior performance compared to state-of-the-art methods, especially under lifelong variations.

**Our contributions** are summarized as follows:

- To address the task gap and challenges posed by lifelong variations, we introduce Class-Relational Label Smoothing (CRLS) that transforms one-hot hard labels into soft labels while considering visual similarities between classes.

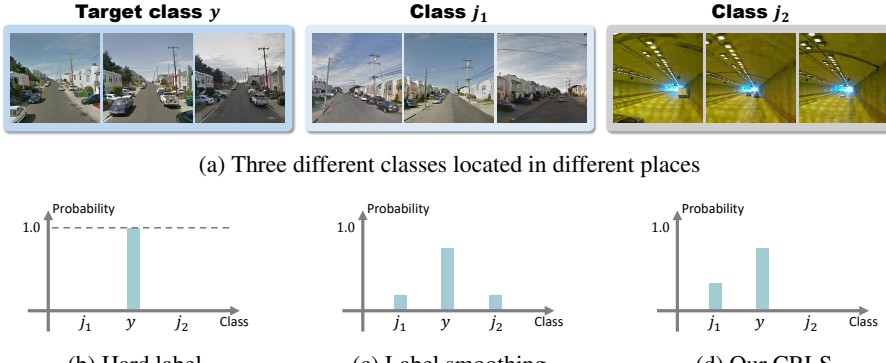

(a) Three different classes located in different places

(b) Hard label       (c) Label smoothing       (d) Our CRLS

Figure 2: **Conceptual comparison between previous and our works.** (a) The target class $y$ shows a street scene with buildings, which is visually similar to class $j_1$, while class $j_2$ shows a markedly different scene inside a tunnel. (b) Hard label strictly assigns the probability to the target class $y$, treating the visually similar class $j_1$ as a negative class. (c) Label smoothing (Szegedy et al., 2016) distributes probability evenly across all classes, including the visually dissimilar class $j_2$ as a positive class. (d) Our CRLS smoothes the label distribution being aware of visual relationships, thus treating class $j_1$ as a as pseudo-positive class and class $j_2$ as a pseudo-negative class.

- Building on the insights from the behavior of class weights during training, we propose Class Stability Weighting (CSW), which dynamically adjust the label smoothing process according to the stability of class weights.

- We organize the lifelong category based on the given temporal span and conduct comprehensive experiments on a diverse set of 17 benchmarks. We achieve state-of-the-art performance across the majority of benchmarks, showing notable improvements on lifelong benchmarks.

## 2 RELATED WORK

**Visual place recognition.** The task of Visual Place Recognition (VPR) has been approached through image retrieval techniques, where the location of a query image is determined by matching to geo-tagged images within an extensive database. Traditionally, this involved the aggregation of hand-crafted features, such as SIFT (Lowe, 2004) and SURF (Bay et al., 2008), for a task of VPR (Knopp et al., 2010; Gronat et al., 2013; Torii et al., 2013a; 2015). With the advent of deep learning, many recent works (Wang et al., 2022; Zhu et al., 2023; Shen et al., 2023; Leyva-Vallina et al., 2023; Hausler et al., 2021; Arandjelovic et al., 2016; Ge et al., 2020; Warburg et al., 2020; Wang et al., 2023; Lu et al.) have exploited metric learning losses, such as contrastive and triplet losses, to get more discriminative image representations. StructVPR (Shen et al., 2023) enhances RGB global features with structural knowledge through segmentation masks and knowledge distillation. $R^2$Former (Zhu et al., 2023) integrates retrieval and reranking with a transformer-based framework, offering a more computationally efficient alternative to other RANSAC-based methods (Hausler et al., 2021; Wang et al., 2022) in the reranking stage. GCL (Leyva-Vallina et al., 2023) extends the traditional contrastive loss in a generalized manner and constructs soft labels based on view overlap, enabling the consideration of continuous relations of viewpoints between images while training VPR models. The methods introduced above are primarily built upon ranking losses, with the use of a mining strategy such as hard negative mining.

More recently, CosPlace (Berton et al., 2022) newly proposes the extensive San Francisco eXtra Large (SF-XL) dataset, showing the scalability issues of the mining strategy in existing VPR methods. To overcome this problem, it leverages a classification loss as a proxy task, specifically Large Margin Cosine Loss (LMCL) (Wang et al., 2018), achieving the superior performances on VPR benchmarks. Furthermore, EigenPlaces (Berton et al., 2023) revises the labeling strategy of the SF-XL dataset to incorporate classes with diverse views, effectively addressing a challenge of substantial viewpoint changes in VPR methods. However, among the VPR methods under the study that utilize classification loss as a proxy task for training, there has been no exploration into the effectiveness of using label smoothing or incorporating class similarity to address the task gap.

**Label smoothing and its retrieval application.** Label smoothing (Szegedy et al., 2016) is first proposed to address the issue of overconfidence in model predictions, thereby enhancing the model's generalization ability with smoother decision boundaries. Since its introduction, it has been widely used for network calibration (Wang, 2023) from many works (Pereyra et al., 2017; Müller et al., 2019; Liu et al., 2022; Park et al., 2023; Liu et al., 2023). MbLS (Liu et al., 2022) applies label smoothing selectively, using the distance over logits and a controllable margin for flexible generalization. Building upon the work, ACLS (Park et al., 2023) further adjusts the label smoothing intensity according to the logit distances. CALS (Liu et al., 2023) is another calibration technique that calculates class-wise penalty weights from the loss using the augmented Lagrangian multiplier method. It adjusts label smoothing intensity based on these penalty weights, enabling class-wise calibration. However, these methods were not proposed for image retrieval or VPR.

For a retrieval application of person re-identification (re-ID), Luo et al. (2019) proposed utilizing the label smoothing as a recipe for establishing a strong baseline. It has since been adopted and modified by several re-ID works to improve search accuracy (Zhu et al., 2020; Cho et al., 2022; Jia et al., 2022). Zhu et al. (2020) introduced an instance-wise adaptive label smoothing that adjusts smoothing strengths based on network predictions and viewpoint variability within each identity class. Another instance-wise label smoothing was proposed by Cho et al. (2022), which applies different smoothing levels to local features considering the relationship between global and local features. However, the existing works including calibration have not considered visual similarity-based class relations to make representation space continuous over visual changes.

## 3 METHODS

### 3.1 PRELIMINARIES

Recent works (Berton et al., 2022; 2023) in VPR utilize a classification loss as a proxy task, employing CosFace (Wang et al., 2018) as the training loss. A margin-based logit function for any class $j$ including a target class $y$ can be formulated with a margin $m$ and a scale factor $s$ as:

$$l\left(\cos\theta_j\right) = \begin{cases} s(\cos\theta_j - m) & j = y \\ s\cos\theta_j & j \neq y \end{cases}. \tag{1}$$

Predicted probability $p_j$ are calculated using cosine similarity and softmax, which is specifically formulated as follows:

$$p_j = \frac{\exp\left(l\left(\cos\theta_j\right)\right)}{\sum_k \exp\left(l\left(\cos\theta_k\right)\right)}, \; \cos\theta_k = \frac{W_k^T \cdot x}{||W_k|| \cdot ||x||}, \tag{2}$$

where $W_k$ is a weight vector of class $k$ and $x$ is a feature vector of an input image. These probabilities are then utilized to compute cross-entropy loss for CosFace loss as follows:

$$\mathcal{L}_{\text{CosFace}}\left(q, p\right) = H\left(q, p\right) = -\sum_k q_k \log\left(p_k\right), \tag{3}$$

where hard target distribution $q_k$ is 1 for the target class and 0 for the rest. Subsequently, the model trained on the classification loss is deployed for a task of image retrieval, specifically in the context of VPR.

Label Smoothing (LS) (Szegedy et al., 2016) is a regularization technique that adjusts the hard target, typically represented as one-hot encoded vectors, towards a smoother distribution. This adjustment encourages the model to be less confident, thereby improving its generalization capabilities. The LS technique modifies the target distribution $q$ according to the smoothing parameter $\alpha$ and the number of classes $K$, as shown in the following equations:

$$f_{\text{LS}}(q_j) = \begin{cases} (1 - \alpha), & j = y, \\ \frac{\alpha}{(K-1)}, & j \neq y, \end{cases} \tag{4}$$

$$\mathcal{L}_{\text{LS}}(q, p) = H(f_{\text{LS}}(q), p).$$

In this work, we argue that assigning equal smoothing across all classes may be less effective for image retrieval task. For the task, establishing a continuous representation space that reflects visual differences is crucial for extreme changes such as lifelong variations, while a discretized representation space is usually sufficient for classification tasks.

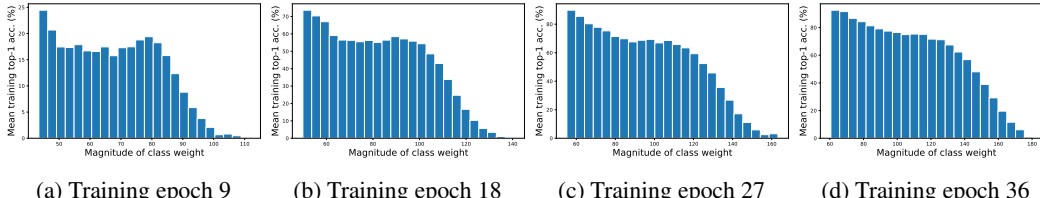

      (a) Training epoch 9      (b) Training epoch 18      (c) Training epoch 27      (d) Training epoch 36

Figure 3: **Relationship between training accuracy and magnitude of class weight.** (a)-(d) show histograms depicting this relationship at different training epochs. The x-axis represents the magnitude of the class weight, defined as the $l_2$-norm of the weight vector, while the y-axis shows mean classification accuracy for classes within the specific magnitude range. It appears that relatively lower magnitudes consistently correspond to higher training accuracy throughout the overall training.

### 3.2 CLASS-RELATIONAL LABEL SMOOTHING

To better reflect the visual differences more within the representation space, we here introduce Class-Relational Label Smoothing (CRLS) extending LS by integrating visual information from inter-class relations. Thanks to the cosine-based classification loss demonstrated in Eq. 2 and 1, the $l_2$-normed class weight vectors are aligned within the same distance space as $l_2$-normed feature vectors, thereby enabling the calculation of visual similarity across classes. Hence, to capture the visual difference between classes, the similarity-based affinity between classes is calculated using the normalized dot product of their class weights $W \in \mathbb{R}^{K \times C}$ of dimensionality $C$, represented as:

$$A_{y,j} = \sigma(W_y)^\top \sigma(W_j), \tag{5}$$

where the $l_2$-normalization function $\sigma(\mathbf{f}) = \mathbf{f}/\|\mathbf{f}\|$, and $W_y$ and $W_j$ are the weights of the classifier for the target class and any other classes, respectively. We then apply a softmax function to the class relations to obtain a label distribution. Additionally, we leveraged a temperature parameter $\tau$ to enhance a contrast in visual similarities among classes, as follows:

$$\hat{A}_{y,j} = \frac{\exp(A_{y,j}/\tau)}{\sum_{k \neq y} \exp(A_{y,k}/\tau)}, \ \ \forall j \in \{1, \dots, K\}. \tag{6}$$

Finally, our loss using CRLS is reformulated with the constructed label distribution $\hat{A}_y$ as follows:

$$f_{\text{CRLS}}(q_j) = \begin{cases} (1 - \alpha), & j = y, \\ \alpha \hat{A}_{y,j}, & j \neq y, \end{cases} \tag{7}$$

$$\mathcal{L}_{\text{CRLS}}(q, p) = H(f_{\text{CRLS}}(q), p).$$

We assign labels in a sequence that mirrors the visual similarity to the target class, thereby enhancing the model's ability to generalize across visually similar scenarios, *e.g.*, lifelong variation.

### 3.3 LOSS INTEGRATION WITH CLASS STABILITY WEIGHTING

Our CRLS approach builds a target distribution $f_{\text{CRLS}}(q)$ using class weights changing during training, which may causes fluctuations in the target distribution during training. These rapid fluctuations can destabilize the learning process. Classes that are more difficult to learn tend to require more updates for their weights during training, which can lead to larger fluctuations. These frequent updates may result in higher weight magnitudes for such classes. Therefore, we investigate a relationship between training accuracy and magnitude of class weight. Figure 3 empirically shows an inversely proportional relationship throughout the entire training epochs; specifically, as the magnitude of class weights increases, we observe a decline in training accuracy. A class achieving high training accuracy tends to exhibit less fluctuation in its weight during training, thus giving a chance for the stable application of CRLS. We therefore define the magnitude as a measure of class stability, and use it for final loss function. We integrate two losses of LS and CRLS to make the training more stable through Class Stability Weighting (CSW) as:

$$\mathcal{L} = \gamma_{y_i} \mathcal{L}_{\text{LS}} + (1 - \gamma_{y_i}) \mathcal{L}_{\text{CRLS}}, \tag{8}$$

where $\gamma_{y_i}$ is the loss weight for input image $i$, determined for each target class $y_i$ by min-max normalization of the magnitudes over all classes. Integrating CSW with CRLS enables dynamic adjustment of each loss's contribution, effectively mitigating potential fluctuations during training.

Table 1: **Overview of VPR benchmarks.** This table provides dataset statistics, including the number of images in the database and queries, and their categorization as lifelong, multi-view, or single-view.

| Dataset | #Database | #Queries | Lifelong | Multi-View | Single-View |
|---|---|---|---|---|---|
| SF-XL test v1 (Berton et al., 2022) | 2.8M | 1000 | ✓ | ✓ | |
| SF-XL test v2 (Berton et al., 2022) | 2.8M | 598 | ✓ | ✓ | |
| MSLS Val (Warburg et al., 2020) | 18.9k | 740 | ✓ | | ✓ |
| MSLS Challenge (Warburg et al., 2020) | 38.7k | 27k | ✓ | | ✓ |
| AmsterTime (Yildiz et al., 2022) | 1231 | 1231 | ✓ | ✓ | |
| Eynsham (Cummins & Newman, 2009) | 23.9k | 23.9k | | ✓ | |
| Pitts30k (Torii et al., 2013b) | 6.8k | 10k | | ✓ | |
| Pitts250k (Torii et al., 2013b) | 8.3k | 84k | | ✓ | |
| Tokyo 24/7 (Torii et al., 2015) | 76k | 315 | | ✓ | |
| San Francisco Landmark (Chen et al., 2011) | 1M | 598 | | ✓ | |
| SVOX Night (Berton et al., 2021) | 17k | 823 | | | ✓ |
| SVOX Overcast (Berton et al., 2021) | 17k | 872 | | | ✓ |
| SVOX Rain (Berton et al., 2021) | 17k | 937 | | | ✓ |
| SVOX Snow (Berton et al., 2021) | 17k | 870 | | | ✓ |
| SVOX Sun (Berton et al., 2021) | 17k | 854 | | | ✓ |
| St Lucia (Milford & Wyeth, 2008) | 1549 | 1464 | | | ✓ |
| Nordland (Sünderhauf et al., 2013) | 27.5k | 27.5k | | | ✓ |

**Gradient analysis on class weight.** We assume that weight magnitude closely reflects the cumulative gradient magnitude over training epochs. By analyzing gradient magnitude, we can understand the relationship between training accuracy and magnitude of class weight. The training accuracy is associated with a similarity to target class, and the similarity reflects the difficulty of a given instance (Meng et al., 2021). The magnitude of the derivative of the loss function with respect to the class weight $W_{y_i}$, where $y_i$ is the target class of an input $i$, can be simplified with respect to $\cos \theta_{y_i}$ as follows[1]:

$$\left\| \frac{\partial L_{CE}}{\partial W_{y_i}} \right\| = \frac{\partial L_{CE}}{\partial l(\cos \theta_{y_i})} \frac{\partial l(\cos \theta_{y_i})}{\partial \cos \theta_{y_i}} \left\| \frac{\partial \cos \theta_{y_i}}{\partial W_{y_i}} \right\|$$

$$\propto (1 - p_{y_i}) \sqrt{1 - \cos^2 \theta_{y_i}} \quad \text{(with respect to } \cos \theta_{y_i} \text{).} \tag{9}$$

The derived equation consists of the product of two terms: $(1 - p_{y_i})$ and $\sqrt{1 - \cos^2 \theta_{y_i}}$. Both of these terms are inversely related to $\cos \theta_{y_i}$, which means $\cos \theta_{y_i}$ controls the emphasis on gradients based on the difficulty during training. The change in class weight while updating tends to depend on the magnitude of the gradient. Consequently, in CSW, the weight magnitude helps to identify saturated classes that require minimal change, which aids in stabilizing the training process.

# 4 EXPERIMENTS

## 4.1 DATASETS AND EVALUATION

We extensively evaluate our method on diverse benchmarks outlined in EigenPlaces (Berton et al., 2023) for comprehensive and fair comparisons. These benchmarks are initially categorized into *Multi-View*, featuring image pairs that include both frontal and lateral perspectives of the road, and *Single-View*, where image pairs consist solely of frontal views relative to the road. Additionally, we introduce a new category termed *Lifelong*, where the benchmark contains datasets with significant temporal variations, capturing changes over extended periods. Among datasets providing temporal span information, we classify those with at least a two-year gap between query and database images as lifelong, allowing for long-term changes in the captured environments. Detailed statistics and the category of each benchmark are shown in Table 1. For fairness and reproducibility, we conduct all experiments using publicly available repositories[2,3] for VPR evaluation.

We summarize the details of five benchmarks in the lifelong category as follows:

**SF-XL test v1 and v2** (Berton et al., 2022). The SF-XL database, shared for evaluations in both test v1 and test v2, encompasses the entire city of San Francisco with 2.8M testing images captured in 2013. The images are sourced from Google StreetView and provide a wide range of challenging

---

[1]The detailed derivation can be found in the Appendix.

[2]https://github.com/gmberton/VPR-methods-evaluation

[3]https://github.com/gmberton/VPR-datasets-downloader

Table 2: **Comparison on the lifelong category.** This table categorizes methods by backbone type into three sections: Others, VGG, and ResNet. The best results are highlighted in bold, and the second best results are underlined, excluding the Others. Recall@1 and Recall@5 (%) are reported.

| Method | Backbone | Dim. | SF-XL test v1 | | SF-XL test v2 | | MSLS Val | | MSLS Challenge | | Amster. | |
|---|---|---|---|---|---|---|---|---|---|---|---|---|
| | | | R@1 | R@5 | R@1 | R@5 | R@1 | R@5 | R@1 | R@5 | R@1 | R@5 |
| TransVPR | - | 256 | 12.0 | 21.0 | 31.8 | 47.0 | 70.9 | 85.0 | 49.0 | 68.7 | 10.6 | 21.1 |
| StructVPR | MobileNetV2 | 448 | - | - | - | - | 83.0 | 91.0 | 64.5 | 80.4 | - | - |
| $R^2$Former | ViT-S | 256 | 20.3 | 31.9 | 37.1 | 60.4 | 80.8 | 90.9 | 56.6 | 75.8 | 12.9 | 26.8 |
| GCL | VGG-16 | 512 | 9.6 | 15.3 | 37.3 | 54.0 | 64.7 | 77.0 | 42.9 | 55.9 | 10.2 | 22.7 |
| CosPlace | VGG-16 | 512 | 65.9 | 75.3 | 83.1 | 91.3 | 82.4 | 90.4 | 61.2 | 73.8 | 38.7 | 61.3 |
| EigenPlaces | VGG-16 | 512 | 69.4 | 78.4 | 86.3 | 93.6 | 84.6 | 90.3 | 60.9 | 72.7 | 38.0 | 59.2 |
| **Ours** | VGG-16 | 512 | 73.5 | 80.4 | 87.0 | 94.1 | 85.3 | 91.5 | 63.4 | 75.1 | 39.2 | 61.8 |
| $R^2$Former | ResNet-50 | 256 | 19.0 | 30.4 | 47.0 | 65.9 | 79.6 | 90.7 | 57.0 | 74.1 | 16.7 | 31.4 |
| MixVPR | ResNet-50 | 512 | 61.2 | 72.0 | 85.6 | 91.8 | 83.5 | 92.0 | 60.0 | 73.6 | 35.7 | 53.0 |
| MixVPR | ResNet-50 | 4096 | 72.5 | 79.3 | 88.6 | 94.5 | 88.4 | 93.5 | 64.3 | 76.5 | 40.8 | 58.9 |
| GCL | ResNet-50 | 2048 | 11.4 | 20.6 | 42.3 | 57.0 | 66.2 | 77.8 | 43.2 | 59.7 | 14.6 | 30.1 |
| CosPlace | ResNet-50 | 2048 | 76.4 | 83.3 | 88.8 | 95.0 | 87.3 | 94.0 | 67.5 | 77.9 | 47.7 | 69.8 |
| EigenPlaces | ResNet-50 | 2048 | 84.1 | 89.1 | 90.8 | 95.7 | 89.1 | 93.8 | 67.9 | 77.7 | 48.9 | 69.5 |
| **Ours** | ResNet-50 | 2048 | 86.0 | 90.4 | 92.3 | 96.0 | 90.1 | 94.1 | 68.8 | 78.9 | 51.1 | 72.5 |

scenarios such as significant viewpoint changes, with highly accurate GPS data. SF-XL test v1 consists of 1,000 query images sourced from Flickr, captured across various years from 2006 to 2020, leading to a temporal gap of up to 14 years. SF-XL test v2 uses a set of 592 queries from from the San Francisco Landmark dataset (Chen et al., 2011), which was released in 2011, ensuring a minimal temporal gap of two years since the SF-XL database images were captured in 2013. Given the frequent modification of urban structures (*e.g.*, buildings), SF-XL provides a realistic test for lifelong scenarios with significant temporal variability.

**MSLS Val and Challenge** (Warburg et al., 2020) is a crowdsourced dataset for lifelong visual place recognition, containing tens of thousands of images from 30 major cities across six continents. For the challenge set, GPS data is withheld to ensure the integrity of evaluations, which are conducted through an online competition platform (Pavao et al., 2023). Notably, it spans seven years of temporal coverage, making it particularly suitable for evaluating lifelong VPR scenarios.

**AmsterTime** (Yildiz et al., 2022) consists of a set of 1,231 grayscale historical images as queries and 1,231 contemporary photos as the gallery in Amsterdam. It contains matching labels between queries and gallery images, confirmed by human experts. This dataset presents one of the most challenging lifelong scenarios, with an extreme temporal gap of *over a century*.

For performance evaluation, we follow the evaluation protocol commonly adopted in previous VPR studies (Arandjelovic et al., 2016; Ge et al., 2020; Warburg et al., 2020; Hausler et al., 2021; Wang et al., 2022; Zhu et al., 2023; Leyva-Vallina et al., 2023). We use a distance threshold of 25 meters to identify positive matches by calculating physical distances from the provided location data, except for AmsterTime (Yildiz et al., 2022) and Nordland (Sünderhauf et al., 2013). AmsterTime provides predefined query-positive pairs, and thus we use the labels directly without distance calculation. For Nordland, which consists of aligned frames from four seasons, a query is considered accurately localized if one of the top-N predictions falls within ten frames of its corresponding ground truth in the database. As is standard in VPR, we utilize recall@K (R@K) as the evaluation metric, measuring the proportion of queries with at least one positive image among the top-K shortlisted results.

## 4.2 IMPLEMENTATION DETAILS

To validate the effectiveness of our method, we employ several deployable architectures, including VGG-16 (Simonyan & Zisserman, 2014), ResNet-50 (He et al., 2016), DINOv2 (Oquab et al., 2023), as backbone networks. For CNN backbones, we extract global descriptors using GeM pooling (Radenović et al., 2018) followed by a fully connected layer. For DINOv2, following Izquierdo & Civera (2024), we apply two-layer MLPs for token embeddings, followed by a fully connected layer for the dimensionality reduction, and fine-tune only the last four blocks. Following the class labeling strategy by EigenPlaces (Berton et al., 2023), we utilize the SF-XL training dataset, cropping a total of 6.72M images from 3.43M panoramas. These images are classified into approximately 263.9k classes and further organized into 18 groups, with each training epoch involving two groups. For training stability, we temporarily freeze our CRLS during the initial 9 epochs as a warm-up, out of a total of 40 epochs. We set the smoothing parameter $\alpha = 0.2$ and

Table 3: **Comparison on the multi-view category.** The layout is the same as in Table 2.

| Method | Backbone | Dim. | Eynsham | | Pitts 30k | | Pitts 250k | | Tokyo 24/7 | | San. Landmark | |
|---|---|---|---|---|---|---|---|---|---|---|---|---|
| | | | R@1 | R@5 | R@1 | R@5 | R@1 | R@5 | R@1 | R@5 | R@1 | R@5 |
| TransVPR | - | 256 | 79.8 | 87.9 | 60.7 | 80.6 | 54.5 | 74.8 | 33.3 | 51.7 | 26.4 | 42.0 |
| StructVPR | MobileNetV2 | 448 | - | - | **85.1** | **92.3** | - | - | - | - | - | - |
| $R^2$Former | ViT-S | 256 | **82.4** | **90.3** | 73.1 | 88.7 | **70.2** | **87.5** | **48.3** | **65.1** | **42.8** | **61.4** |
| GCL | VGG-16 | 512 | 69.0 | 79.4 | 61.7 | 80.1 | 53.4 | 72.5 | 37.1 | 57.5 | 35.8 | 51.3 |
| CosPlace | VGG-16 | 512 | 88.3 | 92.7 | 88.4 | 94.6 | 89.7 | 96.6 | 82.5 | 90.8 | 80.8 | 87.5 |
| EigenPlaces | VGG-16 | 512 | 89.4 | 93.6 | 89.7 | 95.0 | 91.2 | 96.8 | 82.5 | 90.8 | 83.8 | 90.6 |
| **Ours** | VGG-16 | 512 | **89.5** | **93.7** | **90.1** | **95.2** | **91.6** | **96.9** | 81.0 | **92.4** | **84.8** | 90.6 |
| $R^2$Former | ResNet-50 | 256 | 84.9 | 91.3 | 76.5 | 90.3 | 72.5 | 88.1 | 51.7 | 70.2 | 50.5 | 63.5 |
| MixVPR | ResNet-50 | 512 | 87.8 | 92.1 | 90.6 | 95.6 | 93.2 | 98.0 | 79.4 | 88.6 | 79.8 | 86.3 |
| MixVPR | ResNet-50 | 4096 | 89.6 | 93.2 | 91.6 | 95.6 | **94.1** | **98.1** | 86.3 | 91.1 | 84.6 | 90.3 |
| GCL | ResNet-50 | 2048 | 71.3 | 82.1 | 72.0 | 87.5 | 68.0 | 84.4 | 43.2 | 59.7 | 41.3 | 57.0 |
| CosPlace | ResNet-50 | 2048 | 90.0 | 93.9 | 90.9 | 95.7 | 92.3 | 97.4 | 87.3 | 94.0 | 87.1 | 91.1 |
| EigenPlaces | ResNet-50 | 2048 | 90.7 | 94.4 | **92.5** | **96.8** | **94.1** | 97.9 | 92.7 | 96.2 | 89.6 | 94.3 |
| **Ours** | ResNet-50 | 2048 | **90.9** | **94.6** | 92.3 | 96.3 | **94.2** | **98.2** | **94.0** | **96.8** | **91.6** | **95.2** |

Table 4: **Comparison on the single-view category.** The layout is the same as in Table 2. (*) indicates the use of high computational resources. Recall@1 (%) is reported.

| Method | Backbone | Dim. | SVOX Night | SVOX Overcast | SVOX Rain | SVOX Snow | SVOX Sun | St Lucia | Nordland |
|---|---|---|---|---|---|---|---|---|---|
| TransVPR | - | 256 | 6.4 | 61.1 | 26.9 | 47.0 | 13.3 | 81.4 | 22.2 |
| StructVPR | MobileNetV2 | 448 | - | - | - | - | - | - | **56.1** |
| $R^2$Former | ViT-S | 256 | **13.5** | **75.7** | **47.6** | **60.7** | **28.1** | **93.4** | 24.6 |
| GCL | VGG-16 | 512 | 4.4 | 57.2 | 32.4 | 48.0 | 9.0 | 59.1 | 13.3 |
| CosPlace | VGG-16 | 512 | 44.8 | 88.5 | 85.2 | 89.0 | 67.3 | 95.3 | 58.5 |
| EigenPlaces | VGG-16 | 512 | 42.3 | 89.4 | 83.5 | 89.2 | 69.7 | 95.4 | 54.5 |
| **Ours** | VGG-16 | 512 | **47.6** | **91.5** | **85.3** | **90.5** | **70.8** | **96.2** | **59.5** |
| $R^2$Former | ResNet-50 | 256 | 22.4 | 78.1 | 54.4 | 69.8 | 34.2 | 90.0 | 31.9 |
| MixVPR | ResNet-50 | 512 | 45.8 | 93.8 | 86.9 | 93.6 | 79.2 | 99.2 | 66.5 |
| MixVPR | ResNet-50 | 4096* | 62.9 | **96.2** | **92.1** | **97.0** | 85.4 | **99.5** | **76.7** |
| GCL | ResNet-50 | 2048 | 8.4 | 54.5 | 34.4 | 47.0 | 11.0 | 74.8 | 13.9 |
| CosPlace | ResNet-50 | 2048 | 50.7 | 92.2 | 87.0 | 92.0 | 78.5 | **99.6** | 71.8 |
| EigenPlaces | ResNet-50 | 2048 | 58.9 | 93.1 | 90.0 | 93.1 | **86.4** | **99.6** | 71.2 |
| **Ours** | ResNet-50 | 2048 | **64.6** | 94.0 | 90.3 | 94.1 | 85.5 | **99.6** | 73.1 |

temperature $\tau = 0.1$. The model is trained with a batch size of 320 using Adam (Diederik, 2014) optimizer with a learning rate of $1 \times 10^{-4}$. All training is performed on two RTX 3090 GPUs, and a complete training takes approximately 17 hours.

### 4.3 COMPARISON WITH STATE-OF-THE-ARTS

We conduct comprehensive comparisons of our method with recent state-of-the-art methods, including EigenPlaces (Berton et al., 2023), CosPlace (Berton et al., 2022), GCL (Leyva-Vallina et al., 2023), $R^2$Former (Zhu et al., 2023), MixVPR (Ali-Bey et al., 2023), StructVPR (Shen et al., 2023) and TransVPR (Wang et al., 2022). We utilize author-released pre-trained networks for benchmark evaluations. For StructVPR, as its code or pre-trained network has not been publicly released, we reference its performance as reported in the original paper. Both CosPlace and EigenPlaces are trained on the SF-XL dataset. GCL, $R^2$Former, and StructVPR utilize MSLS for training, while MixVPR is trained on the Google StreetView (GSV) (Ali-bey et al., 2022) dataset. We also report the performance of TransVPR, which is trained on MSLS and employs a custom-designed transformer-based backbone. The extensive results are reported in Table 2, 3, 4, representing lifelong, multi-view, and single-view category, respectively.

For the lifelong category in Table 2, which is the primary focus of this paper, our approach consistently achieves state-of-the-art performance across all benchmarks. By effectively utilizing visual relationships across classes, our method successfully handles challenging lifelong variations with superior performance, whereas no single previous method consistently outperforms others in this category. For the multi-view category in Table 3, our results are either comparable to or surpass the current state-of-the-art, with our method achieving better performance in most cases. For the single-view category in Table 4, we report Recall@1 for clarity, with Recall@5 provided in the Appendix. While MixVPR, using ResNet-50 with 4,096 feature dimensions, shows strong results compared to other methods, it requires significantly more computational resources, utilizing 4,096 dimensions versus our method's 2,048 dimensions. Nevertheless, our method demonstrates competitive performance, and notably, we achieve superior results compared to other state-of-the-art methods with the same feature dimensionality. Moreover, MixVPR does not achieve standout performance

Table 5: **Comparison with the methods using foundation models** on the lifelong category. DINOv2 (Oquab et al., 2023) is used as a backbone network. Recall@1 (%) is reported.

| Method | Backbone | Dim. | SF-XL test v1 | SF-XL test v2 | MSLS Val | MSLS Chall. | Amster. | Avg. |
|--------|----------|------|---------------|---------------|----------|-------------|---------|------|
| AnyLoc | DINOv2-G | 49152 | 66.4 | 83.8 | 65.0 | 39.6 | 40.0 | 59.0 |
| SelaVPR | DINOv2-L | 1024 | 55.1 | 72.1 | 87.7 | 69.6 | 36.9 | 64.3 |
| CricaVPR | DINOv2-B | 10752 | 65.8 | 83.3 | 89.1 | 68.1 | 38.6 | 69.0 |
| SALAD | DINOv2-B | 8448 | 88.7 | **94.5** | 92.0 | 75.8 | 58.6 | 81.9 |
| SALAD | DINOv2-B | 2112 | 82.2 | 93.3 | 90.8 | 74.4 | 54.3 | 79.0 |
| **Ours** | DINOv2-B | 2048 | **93.7** | 94.0 | **92.2** | **77.3** | **59.9** | **83.4** |

Table 6: **Ablation study of CRLS and CSW.** We report Recall@1 (%) across five benchmarks in the lifelong category.

| CRLS | CSW | LS | SF-XL test v1 | SF-XL test v2 | MSLS Val | MSLS Chall. | Amster. |
|------|-----|-----|---------------|---------------|----------|-------------|---------|
| | | | 83.8 | 90.6 | 88.4 | 66.3 | 47.8 |
| ✓ | | | 85.0 | 91.1 | 89.5 | 67.8 | 48.8 |
| ✓ | ✓ | | 85.6 | 91.8 | 90.1 | 68.9 | 50.4 |
| ✓ | | ✓ | 85.2 | 90.9 | 89.6 | 67.8 | 48.4 |
| ✓ | ✓ | ✓ | **86.0** | **92.3** | **90.1** | **68.8** | **51.1** |

in the more challenging lifelong and multi-view benchmarks. In summary, our method consistently achieves state-of-the-art or highly competitive performance across all three categories, demonstrating its robustness and superiority compared to recent state-of-the-art approaches.

Given recent advancements in VPR through the discriminative power of foundation models, we further evaluate our method built on DINOv2 (Oquab et al., 2023) as the backbone, with the results presented in Table 5. We compare our method with other foundation model-based methods, including AnyLoc (Keetha et al., 2023), SelaVPR (Lu et al.), CricaVPR (Lu et al., 2024), and SALAD (Izquierdo & Civera, 2024). Anyloc is a zero-shot method without fine-tuning. For SALAD, with a dimensionality of 2,112, we reproduce the method using the author-released code and provided parameters. The experimental results show that our method, based on DINOv2-B with a compact dimensionality of 2,048, achieves state-of-the-art performance across the lifelong benchmarks. Specifically, our method outperforms all others in terms of average R@1, achieving 83.4% across the five lifelong datasets. It surpasses methods using larger models (*e.g.*, DINOv2-L,G) and higher dimensionalities (*e.g.*, 8,448 for SALAD). Notably, on the challenging SF-XL test v1, our method achieves a substantial 5% improvement over the second-best, SALAD.

## 4.4 ABLATION STUDY

**Effectiveness of CRLS and CSW.** We explore the impact of CRLS, CSW, and naïve LS on retrieval performance in Table 6. As a baseline, we use the CosFace loss with one-hot encoded hard labels as EigenPlaces. To solely assess the effectiveness of each component, we employ the vanilla $\mathcal{L}_{\text{CosFace}}$ for cases without LS. Specifically, the loss of CRLS with only CSW is calculated as $\mathcal{L} = \gamma_{y_i}\mathcal{L}_{\text{CosFace}} + (1 - \gamma_{y_i})\mathcal{L}_{\text{CRLS}}$. These experiments are conducted using a ResNet-50 architecture with a feature dimensionality of 2048. We adopt five evaluation datasets that reflect lifelong scenarios: SF-XL test v1, SF-XL test v2, MSLS Val, MSLS Challenge, and AmsterTime. Table 6 demonstrates that while using CRLS alone does provide a performance boost over the baseline, it achieves better results when combined with CSW. This can be attributed to the fact that CRLS is more effective for classes with low weight magnitudes, as these classes tend to exhibit less fluctuation during training. The best performance is obtained under our final loss formulation with LS, where LS appears to compensate for the classes having high magnitudes.

To further evaluate the impact of CSW on CRLS, we apply the computed CSW weights in a reversed manner (e.g., $\gamma_{y_i} \rightarrow 1 - \gamma_{y_i}$). This reversal assigns higher weights to classes with high magnitudes, which are more prone to fluctuations during training. The performance degradation observed in Table 8 demonstrates that the contrapositive approach is harmful to the training process. Conversely, this result validates the design choice implemented in CSW, where weights inversely proportional to the magnitude are assigned to stabilize CRLS.

Table 8: Experiments on contrapositive approach of CSW. R-CSW refers to the reverse CSW. Recall@1 (%) is reported.

| Method | MSLS Val | MSLS Chall. | Amster. |
|--------|----------|-------------|---------|
| Ours | **90.1** | **68.8** | **51.1** |
| + R-CSW | 88.9 | 66.6 | 48.5 |

Table 7: **Ablation study of label smoothing strategies** used as substitutes for CRLS in our training.

| Method | SF-XL test v1 | SF-XL test v2 | MSLS Val | MSLS Chall. | Amster. |
|---|---|---|---|---|---|
| LR (Lienen & Hüllermeier, 2021) | 84.0 | 91.0 | 88.9 | 67.9 | 48.6 |
| ACLS (Park et al., 2023) | 82.2 | 91.1 | 87.7 | 67.2 | 47.4 |
| **CRLS (Ours)** | **86.0** | **92.3** | **90.1** | **68.8** | **51.1** |

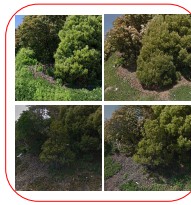 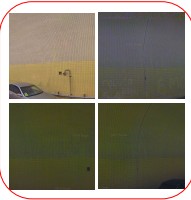 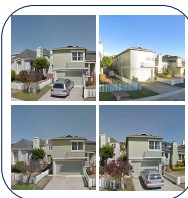 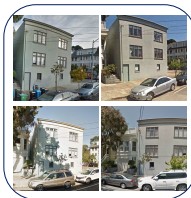

(a) Images within two highest-norm classes      (b) Images within two lowest-norm classes

Figure 4: **Examples of classes according to magnitude.** (a) tends to be hard to be trained and less informative, while (b) shows pleasant classes for appropriate training.

**Comparison with other label smoothing strategies.** To further validate the effectiveness of CRLS, we substitute it with alternative techniques in our final loss formulation and evaluate their performance. Table 7 presents the results of this experiment, where we compare against LR (Lienen & Hüllermeier, 2021) and ACLS (Park et al., 2023), both variants of label smoothing designed for network calibration. The results demonstrate that CRLS consistently outperforms these alternatives. While the other techniques are optimized primarily for network calibration, CRLS is specifically tailored for VPR, leveraging visual relationships between classes to learn a continuous representation space for handling diverse appearance variations.

**Qualitative analysis on the class weight magnitude.** Fig. 4 illustrates that classes with the highest magnitude tend to be less informative, while those with the lowest magnitude are more suitable for training[4]. This distinction in weight magnitudes reflects diverse levels of informativeness and trainability among the training classes. These properties of weight magnitude could be diversely applied in robust training, such as in curriculum learning. Additional qualitative results such as CRLS and retrieval results are presented in Appendix.

## 5 CONCLUSIONS

In this paper, we introduced a novel technique, CRLS, that effectively bridges the task gap between classification and retrieval in visual place recognition, which is particularly vulnerable to extreme visual changes such as lifelong variations. We further enhanced CRLS with CSW, a method that dynamically adjusts the influence of CRLS based on the stability of class weights, quantified by their magnitudes. Our findings, supported by derivative analysis, suggest that the magnitude of class weights serves as an indicator of class stability.

We evaluated our approach through extensive experiments on 17 diverse benchmarks, covering a wide range of scenarios including lifelong, multi-view, and single-view settings. In the majority of cases, our method outperformed existing state-of-the-art methods, while in the remaining cases, it achieved highly competitive performance. Particularly in the most challenging lifelong benchmarks, our approach demonstrated state-of-the-art performance with substantial improvements over existing methods. Furthermore, our method showed its effectiveness in leveraging foundation models, such as DINOv2, for VPR. By integrating ours with DINOv2 backbone, we achieved superior performance compared to other methods utilizing the same backbone, while maintaining a compact feature representation. These results show the robustness and adaptability of our method in tackling lifelong challenges in real-world environments, and in various backbones.

Interestingly, our analysis revealed that class weight magnitudes indicate the informativeness and trainability of classes, which could potentially be utilized in robust training techniques such as curriculum learning. Unfortunately, our analysis is based on the CosFace loss function, while our analytical approach may facilitate such extensions to other losses.

---

[4]More examples can be found in the Appendix

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

## APPENDIX

## A    DERIVATION OF CLASS WEIGHT

The derivative of $L_{CE}$ w.r.t $W_{y_i}$ is computed by chain rule as follows:

$$\frac{\partial L_{CE}}{\partial W_{y_i}} = \frac{\partial L_{CE}}{\partial l(\cos\theta_{y_i})} \frac{\partial l(\cos\theta_{y_i})}{\partial \cos\theta_{y_i}} \frac{\partial \cos\theta_{y_i}}{\partial W_{y_i}}. \tag{10}$$

The derivatives w.r.t logits and cosine similarity can be computed as:

$$\frac{\partial L_{CE}}{\partial l(\cos\theta_{y_i})} = p_{y_i} - 1, \quad \frac{\partial l(\cos\theta_{y_i})}{\partial \cos\theta_{y_i}} = s. \tag{11}$$

By the quotient rule, the derivative of cosine similarity w.r.t the class weight is:

$$\frac{\partial \cos\theta_{y_i}}{\partial W_{y_i}} = \frac{\partial}{\partial W_{y_i}} \left( \frac{W_{y_i}^T x_i}{\|W_{y_i}\|\|x_i\|} \right) = \frac{\|W_{y_i}\|\|x_i\|x_i - \frac{W_{y_i}}{\|W_{y_i}\|}\|x_i\|W_{y_i}^T x_i}{(\|W_{y_i}\|\|x_i\|)^2}$$

$$= \frac{1}{\|W_{y_i}\|} \left( \frac{x_i}{\|x_i\|} - \cos\theta_{y_i}\frac{W_{y_i}}{\|W_{y_i}\|} \right). \tag{12}$$

Incidentally, the gradient vector results in a tangent vector at a point $W_{y_i}$ on a unit sphere, as $\left( \frac{x_i}{\|x_i\|} - \cos\theta_{y_i}\frac{W_{y_i}}{\|W_{y_i}\|} \right) \cdot \frac{W_{y_i}}{\|W_{y_i}\|} = 0$. Since the terms in Eq. 11 are constants, to compute the magnitude of the derivative, we calculate the norm of $\frac{\partial \cos\theta_{y_i}}{\partial W_{y_i}}$ as follows:

$$\left\| \frac{\partial \cos\theta_{y_i}}{\partial W_{y_i}} \right\| = \frac{1}{\|W_{y_i}\|} \sqrt{ \left( \frac{x_i}{\|x_i\|} - \cos\theta_{y_i}\frac{W_{y_i}}{\|W_{y_i}\|} \right)^T \left( \frac{x_i}{\|x_i\|} - \cos\theta_{y_i}\frac{W_{y_i}}{\|W_{y_i}\|} \right) }$$

$$= \frac{1}{\|W_{y_i}\|} \sqrt{1 - \cos^2\theta_{y_i}}. \tag{13}$$

Finally, we can express the magnitude of the derivative of the loss function w.r.t the class weight as:

$$\left\| \frac{\partial L_{CE}}{\partial W_{y_i}} \right\| = \frac{\partial L_{CE}}{\partial l(\cos\theta_{y_i})} \frac{\partial l(\cos\theta_{y_i})}{\partial \cos\theta_{y_i}} \left\| \frac{\partial \cos\theta_{y_i}}{\partial W_{y_i}} \right\|$$

$$= (1 - p_{y_i}) \sqrt{1 - \cos^2\theta_{y_i}} \frac{s}{\|W_{y_i}\|} \quad \because p_{y_i} \in [0, 1], \tag{14}$$

where the overwhelming majority of $\cos\theta_{y_i}$ are greater than zero, as empirically shown in Fig. 5.

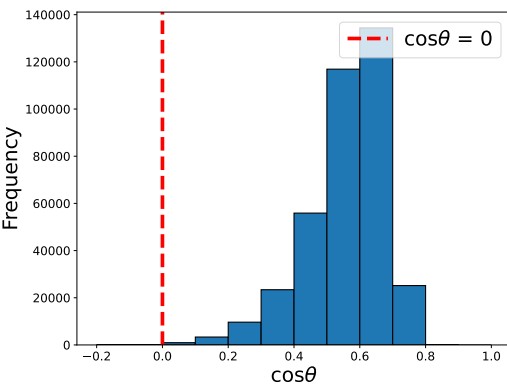

Figure 5: Histogram of $\cos\theta_{y_i}$ values calculated for all input $i$, with the vertical red dashed line indicating $\cos\theta_{y_i} = 0$.

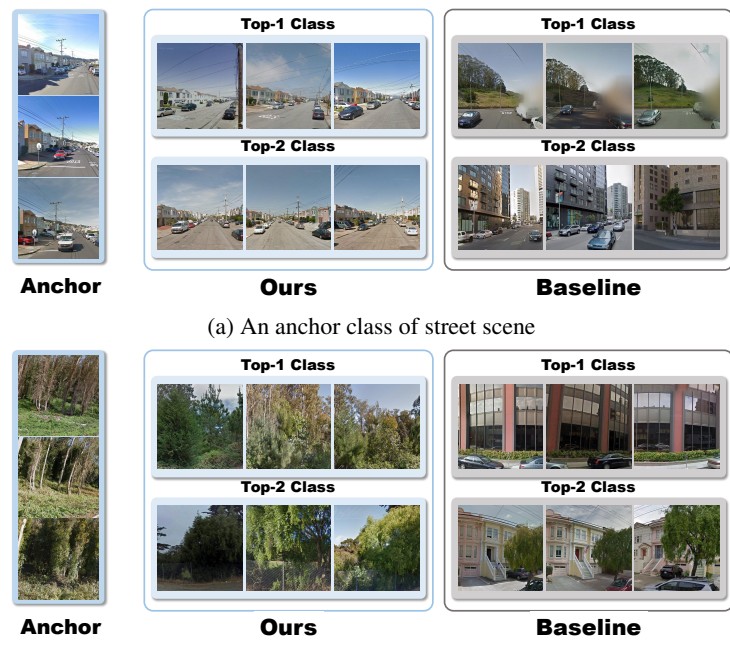

(a) An anchor class of street scene

(b) An anchor class of forest scene

Figure 6: **Qualitative analysis of the impact by CRLS.** The class weights are considered as embedding features, with the class weight of the anchor class treated as the query feature and the weights of other classes treated as database features. The results of top-1, 2 are obtained through the execution of the retrieval procedure at the class level.

Table 9: **Comparison on the single-view category.** The layout is the same as in Table 2. (*) indicates the use of high computational resources. Recall@5 (%) is reported.

| Method | Backbone | Dim. | SVOX Night | SVOX Overcast | SVOX Rain | SVOX Snow | SVOX Sun | St Lucia | Nordland |
|---|---|---|---|---|---|---|---|---|---|
| TransVPR | - | 256 | 15.2 | 80.5 | 49.3 | 72.0 | 29.2 | 90.4 | 37.6 |
| StructVPR | MobileNetV2 | 448 | - | - | - | - | - | - | **75.5** |
| $R^2$Former | ViT-S | 256 | **30.4** | **91.2** | **68.8** | **83.8** | **46.6** | **98.1** | 38.8 |
| GCL | VGG-16 | 512 | 12.8 | 74.5 | 46.9 | 69.4 | 15.9 | 75.3 | 22.6 |
| CosPlace | VGG-16 | 512 | 63.5 | 93.9 | 91.7 | 94.0 | 79.2 | 98.1 | 73.7 |
| EigenPlaces | VGG-16 | 512 | 61.0 | 94.4 | 91.6 | 94.4 | **82.2** | 98.3 | 70.1 |
| **Ours** | VGG-16 | 512 | **66.8** | **96.1** | **93.5** | **95.1** | 81.4 | **98.7** | **74.9** |
| $R^2$Former | ResNet-50 | 256 | 43.4 | 92.9 | 73.7 | 88.5 | 55.6 | 95.5 | 48.6 |
| MixVPR | ResNet-50 | 512 | 62.7 | 97.8 | 93.8 | 97.6 | 90.7 | 99.9 | 80.8 |
| MixVPR | ResNet-50 | 4096* | 79.8 | **98.2** | **96.8** | 98.3 | 93.0 | **100.0** | **87.1** |
| GCL | ResNet-50 | 2048 | 16.4 | 74.2 | 56.1 | 67.7 | 25.6 | 86.6 | 24.7 |
| CosPlace | ResNet-50 | 2048 | 67.4 | 97.7 | 95.1 | 98.4 | 89.7 | 99.9 | 83.8 |
| EigenPlaces | ResNet-50 | 2048 | 76.9 | 97.9 | 96.4 | 97.6 | **95.0** | 99.9 | 83.8 |
| **Ours** | ResNet-50 | 2048 | **83.0** | 98.0 | 96.5 | **98.5** | 94.6 | 99.9 | 84.3 |

Table 10: **Comparison with the methods using foundation models** on the lifelong category. DINOv2 (Oquab et al., 2023) is employed as a backbone network. Recall@5 (%) is reported.

| Method | Backbone | Dim. | SF-XL test v1 | SF-XL test v2 | MSLS Val | MSLS Chall. | Amster. |
|---|---|---|---|---|---|---|---|
| AnyLoc | DINOv2-G | 49152 | 78.1 | 92.3 | 75.4 | 53.1 | 63.8 |
| SelaVPR | DINOv2-L | 1024 | 68.1 | 87.1 | 95.8 | 86.9 | 59.8 |
| CricaVPR | DINOv2-B | 10752 | 76.5 | 91.0 | 95.0 | 80.0 | 60.0 |
| SALAD | DINOv2-B | 8448 | 93.5 | 97.4 | **96.2** | **89.2** | 78.9 |
| SALAD | DINOv2-B | 2112 | 89.4 | 97.3 | 96.1 | 87.1 | 75.1 |
| **Ours** | DINOv2-B | 2048 | **96.6** | **97.5** | 95.7 | 87.9 | **80.7** |

Table 11: **Ablation studies on the hyper-parameters.** We report Recall@1 (%) across five benchmarks in the lifelong category.

(a) Experiments on the smoothing intensity $\alpha$ used in LS and CRLS.

| $\alpha$ | SF-XL test v1 | SF-XL test v2 | MSLS Val | MSLS Chall. | Amster. |
|---|---|---|---|---|---|
| 0.0 | 83.8 | 90.6 | 88.4 | 66.3 | 47.8 |
| 0.1 | 85.1 | 91.6 | 89.2 | 68.1 | 49.4 |
| 0.2 | **86.0** | **92.3** | **90.1** | **68.8** | 51.1 |
| 0.3 | 84.5 | 91.3 | 89.2 | 68.7 | **51.6** |

(b) Experiments on the temperature parameter $1/\tau$.

| $1/\tau$ | SF-XL test v1 | SF-XL test v2 | MSLS Val | MSLS Chall. | Amster. |
|---|---|---|---|---|---|
| 0 | 85.1 | 91.1 | 88.7 | 67.8 | 50.1 |
| 10 | **86.0** | **92.3** | 90.1 | **68.8** | **51.1** |
| 20 | 84.6 | 91.8 | **90.3** | 68.7 | 49.8 |
| 30 | 84.9 | 91.5 | 88.2 | 68.7 | 49.5 |

## B  FURTHER QUALITATIVE AND QUANTITATIVE RESULTS

We prepared examples of the lifelong scenarios and retrieval results to show the effectiveness of our approach. As shown in Fig. 7, AmsterTime uses historical images as queries, which leads to a scenario where buildings in the query and positive images have undergone extreme changes. For EigenPlace, which is trained using classification loss, a goal of the loss is to find an exactly same object. This makes EigenPlace hard to find the matching pairs with extreme changes. In contrast, our method learns in a visually similar-aware manner, and thus the lifelong examples can be handled.

To further validate the effectiveness of CRLS, we conduct retrieval at the class level using class weights in Fig. 6. Upon comparison with the baseline, our method demonstrates that our class weights on representation space better capture semantic or visually-similar information while considering visual differences.

We provide additional quantitative results to further demonstrate the effectiveness of our method. Table 9 presents the Recall@5 performance on the single-view category benchmarks, complementing the Recall@1 results shown in the main paper. Similarly, Table 10 reports the Recall@5 performance of our method and other approaches utilizing the DINOv2 foundation model on the lifelong category benchmarks, providing a more comprehensive evaluation of their performance.

## C  ABLATIONS ON HYPER-PARAMETERS

In addition, we study varying the hyper-parameter $\alpha$, representing the intensity of smoothing as shown in Table 11a. The ablation study finds that $\alpha = 0.2$ delivers the best performances in most cases, validating its selection as the optimal setting for our experiments. We also examine the effect of the temperature parameter $\tau$ in our CRLS approach, with results shown in Table 11b. The experiments reveal that a value of $1/\tau = 10$ ($\tau = 0.1$) generally yields the best performance across the five lifelong datasets, providing an optimal balance.

## D  CLASSIFICATION-BASED APPROACHES IN VISUAL GEO-LOCALIZATION

Contrasting with retrieval-based approaches, another significant area of research in Visual Place Recognition (VPR) is focused on classification-based approaches Weyand et al. (2016); Vivanco Cepeda et al. (2023); Izbicki et al. (2020); Pramanick et al. (2022); Trivigno et al. (2023); Kordopatis-Zilos et al. (2021). Unlike retrieval methods that strive to match a query image with a large database of reference images, classification-based approaches divide the geographic area into discrete cells or regions, with each cell treated as a separate class. This framework transforms the task into a classification problem, where the objective is to identify the correct geographic cell for a given query image. These methods are advantageous for their faster inference times, as they bypass the need for extensive similarity searches required by retrieval-based methods. However, despite the efficiency of classification-based methods, retrieval methods can leverage the fine-grained similarities between

query and database images, enabling more precise localization. Moreover, they are not limited by predefined classes and can potentially localize images at any location covered in the database.

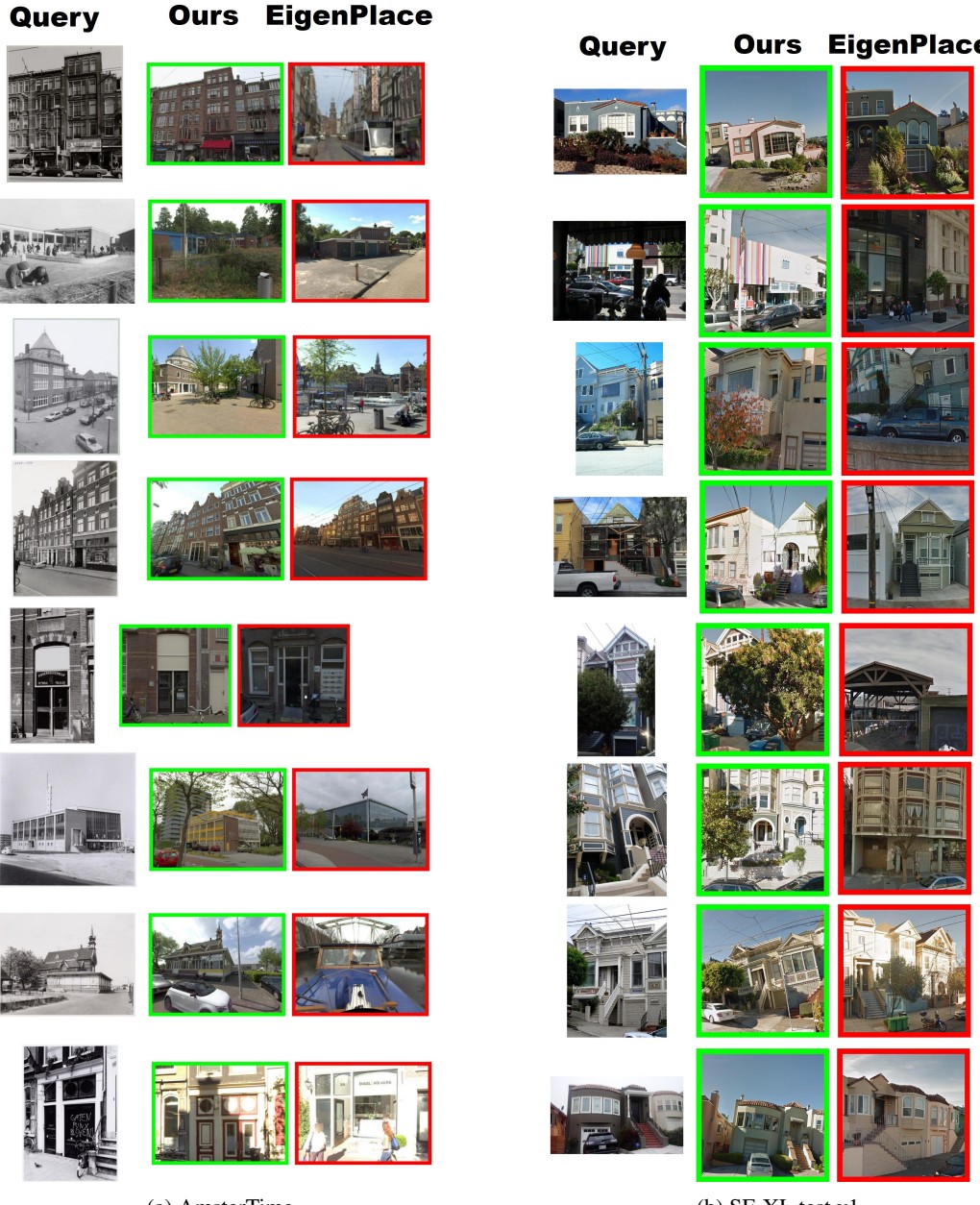

(a) AmsterTime

(b) SF-XL test v1

Figure 7: **Examples of lifelong scenarios and their top-1 retrieval results** from the AmsterTime and SF-XL test v1 datasets. The query and positive images demonstrate significant changes in structure and appearance due to building remodeling over time. We compare the retrieval results of our method with those of EigenPlace, highlighting our method's ability to handle these challenging scenarios effectively.

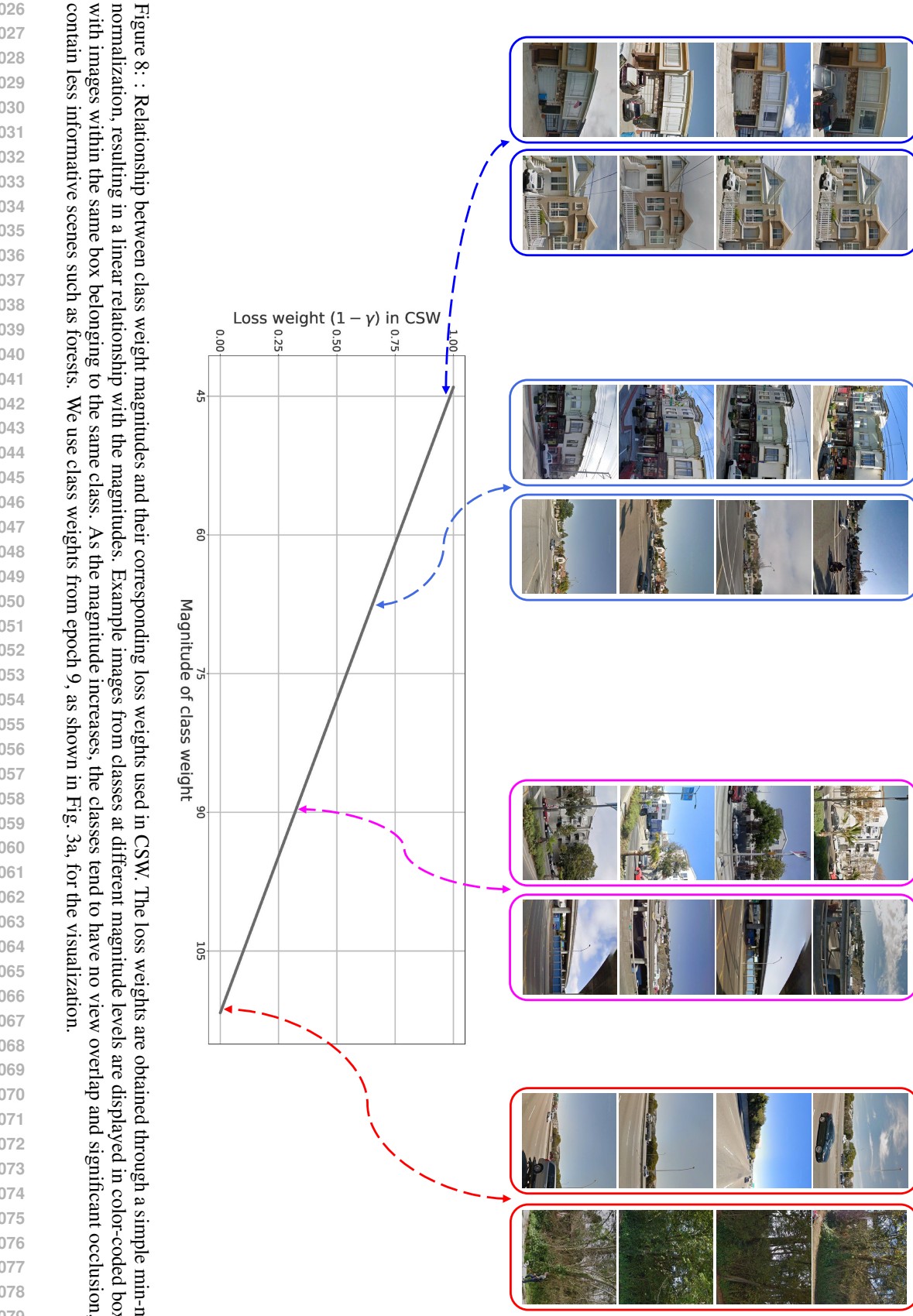

Figure 8: : Relationship between class weight magnitudes and their corresponding loss weights used in CSW. The loss weights are obtained through a simple min-max normalization, resulting in a linear relationship with the magnitudes. Example images from classes at different magnitude levels are displayed in color-coded boxes, with images within the same box belonging to the same class. As the magnitude increases, the classes tend to have no view overlap and significant occlusion, or contain less informative scenes such as forests. We use class weights from epoch 9, as shown in Fig. 3a, for the visualization.

