

| Class cluster 1 | Class cluster 2 | Class cluster 3 |
|---|---|---|

| (a) EigenPlaces (hard label) | (b) Label smoothing | (c) CRLS (Ours) |
|---|---|---|

Figure 1: The t-SNE (Van der Maaten & Hinton, 2008) visualization of the feature space trained with (a) one-hot hard labels (*i.e.*, EigenPlaces), (b) vanilla label smoothing, and (c) our Class-Relational Label Smoothing (CRLS). Different bounding box colors represent different classes, and each class cluster in the top row illustrates two different locations with visually similar buildings. (a) Training with hard labels results in a sparse and discrete feature space, where embeddings are overfitted to specific visual appearances of each class. (b) Label smoothing can partly alleviate the sparsity of hard-label training by softening class boundaries, but it adjusts the label distribution blindly without considering the characteristics of each class. As a result, while the feature space becomes more continuous, it fails to capture the visual continuity required for lifelong variations. (c) Our CRLS overcome this issue by learning the feature space that successfully reflects the differences of the visual appearances between classes. The resulting class clusters exhibit that similar classes in each cluster are closely located, while discriminating visually different classes.

Table 1: **Comparison on the lifelong category.** DINOv2 (Oquab et al., 2023) is employed as a backbone network. Recall@1 (%) is reported.

| Method | Backbone | Dim. | SF-XL test v1 | SF-XL test v2 | MSLS Val | MSLS Chall. | Amster. |
|---|---|---|---|---|---|---|---|
| SALAD | DINOv2-B | 8448 | 88.7 | 94.5 | 92.0 | 75.8 | 58.6 |
| SALAD | DINOv2-B | 2112 | 82.2 | 93.3 | 90.8 | 74.4 | 54.3 |
| BoQ (Ali-bey et al., 2024) | DINOv2-B | 12288 | - | - | **93.8** | **79.5** | 62.9 |
| Ours | DINOv2-B | 2048 | 93.7 | 94.0 | 92.2 | 77.3 | 59.9 |
| Ours | DINOv2-B | 4096 | **94.9** | 94.3 | 92.3 | 78.6 | 62.1 |
| Ours | DINOv2-B | 8192 | 93.6 | **95.2** | 92.6 | 78.6 | **63.6** |

Table 2: **Comparison on the multi-view category.** DINOv2 (Oquab et al., 2023) is employed as a backbone network. Recall@1 (%) is reported.

| Method | Backbone | Dim. | Eynsham | Pitts30k | Pitts250k | Tokyo 24/7 | San. Landmark |
|---|---|---|---|---|---|---|---|
| SALAD | DINOv2-B | 8448 | 91.6 | 92.3 | 95.0 | 94.6 | **92.6** |
| SALAD | DINOv2-B | 2112 | 91.2 | 91.1 | 93.7 | 93.0 | 91.6 |
| BoQ | DINOv2-B | 12288 | 92.1 | 93.7 | 96.6 | 96.5 | - |
| Ours | DINOv2-B | 2048 | 91.8 | 94.0 | 96.6 | 96.5 | 91.5 |
| Ours | DINOv2-B | 4096 | **92.2** | **94.5** | 96.7 | 97.1 | 91.6 |
| Ours | DINOv2-B | 8192 | 92.1 | 94.5 | **97.1** | **97.5** | 90.8 |

Table 3: **Comparison on the single-view category.** DINOv2 (Oquab et al., 2023) is employed as a backbone network. Recall@1 (%) is reported. †uses a two-stage re-ranking.

| Method | Backbone | Dim. | SVOX Night | SVOX Overcast | SVOX Rain | SVOX Snow | SVOX Sun | St Lucia | Nordland |
|---|---|---|---|---|---|---|---|---|---|
| SelaVPR† | DINOv2-L | - | 89.9 | 96.9 | 94.9 | 96.7 | 91.2 | 99.9 | 87.3 |
| SelaVPR | DINOv2-L | 1024 | 73.6 | 92.7 | 86.4 | 92.2 | 77.6 | 99.4 | 69.3 |
| SALAD | DINOv2-B | 8448 | 95.9 | 98.2 | 98.6 | 98.7 | 97.1 | 100.0 | 86.6 |
| SALAD | DINOv2-B | 2112 | 94.5 | 97.7 | 97.4 | 98.2 | 96.3 | 99.9 | 77.8 |
| BoQ | DINOv2-B | 12288 | 97.6 | 98.4 | **98.6** | **99.3** | 97.6 | 99.9 | 90.7 |
| Ours | DINOv2-B | 2048 | 96.2 | 98.2 | 97.7 | 99.0 | 95.8 | 99.9 | **95.0** |
| Ours | DINOv2-B | 4096 | **97.9** | **98.5** | 97.9 | 99.0 | **97.7** | **100.0** | 93.7 |
| Ours | DINOv2-B | 8192 | 97.2 | 98.2 | 97.9 | 99.1 | 97.4 | 100.0 | 93.2 |

Table 4: **Ablation on losses using the DINOv2-B backbone.** Recall@1 and Recall@5 (%) are reported. LS means a naïve label smoothing without CRLS.

| Method | Dim. | SF-XL test v1 | SF-XL test v2 | MSLS Val | MSLS Chall. | Amster. |
|---|---|---|---|---|---|---|
| LS | 2048 | 91.1 | 93.3 | 91.2 | 76.5 | 59.1 |
| Ours | 2048 | **93.7** | **94.0** | **92.2** | **77.3** | **59.9** |