# OpenReview forum: "Class-Relational Label Smoothing for Lifelong Visual Place Recognition"
_ICLR.cc/2025/Conference — ICLR 2025 Conference Withdrawn Submission_

### Official Review · Reviewer_MmfN · 2024-11-01

**Soundness:** 3
**Presentation:** 3
**Contribution:** 2
**Rating:** 6
**Confidence:** 4

**Summary:**

This work proposes to use class-specific soft target labels for classification, applied to visual place recognition and in particular lifelong scenarios, where scene appearance evolves over time while labels remain the same. In the proposed Class-Relational Label Smoothing (CRLS), each ground truth one-hot target label is replaced by a soft one, obtained by pairwise class similarities, as expressed by class weight vectors while the classifier is being learned. In Class Stability Weighting (CSW), a convex combination of the proposed loss and a baseline loss (e.g. label smoothing) is used. The weight of the baseline loss is the norm of the ground truth class weight vector. The main competitor, both in motivation and in comparisons, is label smoothing, where the distribution over background classes is uniform.

**Strengths:**

The paper is well written and easy to follow. The idea is sound and simple. There are only two hyper-parameters and a related ablation study. The method is competitive in most experiments and outperforms competing methods in lifelong visual place recognition.

**Weaknesses:**

The related work, background and competitors in experiments are insufficient. The positioning of providing an improvement over a naive baseline and then competing methods that are not relevant as ideas is problematic. Details follow.

1. CRLS is presented as an improvement over label smoothing, applied to visual place recognition and compared with state of the art (SOTA) methods in the field. The main idea is that the appearance of the input examples is not consistent with the labels, hence the labels are modified. This happens in the particular problem because of evolution over time in the lifelong scenario and the competing SOTA methods are presumably not using any such label modification, although this is not clarified.

    This kind of positioning is problematic, because it treats visual place recognition as a benchmark, with the objective to beat the competition in the benchmark using ideas from anywhere. The proper positioning would be to beat relevant ideas. It would be to provide an improvement over other methods where examples are not consistent with labels, hence discuss such methods in related work and background, then compare with them as competitors in experiments, with visual place recognition being merely the application. Comparison is SOTA in the field of visual place recognition would be a secondary target.

    In this sense, there are several tasks where examples are not consistent with labels, for example learning with weak supervision where labels are noisy, semi-supervised learning where labels are missing, or active learning where examples are selected for labeling. Taking learning with noisy labels as an example, here are some prior works with relevant ideas:

    > [*1] Reed et al. 2014, Training Deep Neural Networks on Noisy Labels with Bootstrapping
    >
    > [*2] Sukhbaatar et al. 2014, Training Convolutional Networks with Noisy Labels
    >
    > [*3] Lee et al. 2017, CleanNet: Transfer Learning for Scalable Image Classifier Training with Label Noise
    >
    > [*4] Li et al. 2017, Learning From Noisy Labels with Distillation
    >
    > [*5] Sharma et al. 2020, NoiseRank: Unsupervised Label Noise Reduction with Dependence Models
    >
    > [*6] Han et al. 2019, Deep Self-Learning From Noisy Labels
    >
    > [*7] Iscen et al. 2022, Learning with Neighbor Consistency for Noisy Labels

    By studying such methods, one realizes that label smoothing is only a very naive baseline. There are many more advanced ideas and some are similar to the proposed method. A quite universal idea is distillation where soft target labels are obtained by a model, possibly the same while being learned. This is used in [*1], [*4] and [*6]. Of those, [*4] also uses a "knowledge graph". This graph plays the same role as the similarity-based affinity in this work, it is just constructed based on class semantics. Beyond pairwise class similarities, [*3] and [*5] use similarities between examples and class prototypes, hence can provide more fine-grained operations per example, while [*7] uses pairwise similarities between examples directly, hence provide even more localized operations per example. The idea of multiple class prototypes would also work well against evolution over time in the lifelong scenario, and there are several such methods in deep metric learning.

    The authors should present such methods as background, also methods from other relevant tasks, then compare with them as competitors. Here, they do not even compare with label smoothing (LS). This could have happened in table 6 for example, which is ablation, but LS is not presented in the absence of CRLS and CSW. There is comparison with other label smoothing strategies (LR and ACLS) in Table 7, but this is treated as ablation study. Instead, more advanced competitors should be chosen and this should be the main comparison. In this comparison, the authors should build on top of the best performing method on visual place recognition and show that adding CRLS works better than adding any other competitor. By doing so, the resulting solution would still beat the SOTA on visual place recognition.

2. For CSW, there are again many strategies by which to assign weights to different loss components based on some sort of confidence, and these weights can be per class like this work or per example, which is more fine-grained. Such ideas can be found in the tasks of semi-supervised learning or active learning. The authors should again study the literature and come up with strong competitors, present them mas background and compare with them.

### Post-rebuttal

Let me first present a connection between CRLS and standard distillation that I sadly realized only after rebuttal and discussion.

In eq. (5), let us replace $W_y$ with $x'$, the feature vector of the input image obtained by a teacher model, and let $W_j$ represent the class weights of the same teacher model. In eq. (7), let us replace $(1-\alpha)$ of case $j=y$ by $(1-\alpha) + \hat{A}_{y,j}$. Then, the CRLS loss is the same as $(1-\alpha)$ times the standard classification loss plus $\alpha$ times the standard distillation loss. Thus, CRLS can be seen as a more efficient version of standard distillation in the sense that we represent an image with its ground truth class weight instead of forwarding it through the teacher to get its soft targets, and parameter $\alpha$ controls the trade-off between supervision and distillation (consistency, or self-supervision).

In light of this observation, I am not sure if the comparison to mean teacher (MT) (Tarvainen and Valpola, 2018) performed towards the end of rebuttal is careful enough to highlight the difference to distillation. This is because MT is using MSE loss and it is unclear how classification and consistency losses are weighted. A careful comparison according to the above would reveal if CRLS has similar effect with distillation, thus providing a cheap alternative, also targeting more clustered embedding spaces (because soft targets are all the same for images of the same class).

Concerning the introduction of the class affinity matrix $A$ in the distillation process, it is worth noting that Li et al. (2017), mentioned in my review, is doing the same, thus being equally a cheap alternative of distillation (applied to learning with noisy labels), although the matrix is based on semantic rather than visual similarity. This connection is important when claiming contribution.

Now, I believe a main concern of all reviewers is how to justify that the proposed approach is really connected with the problem of lifelong visual place recognition (VPR). Here, a difficulty is that the challenge of lifelong VPR is not precisely defined. For example, how is a construction in front of a building different from occlusion? Hence, it is equally difficult for authors to argue that the approach is meant for lifelong VPR as for reviewers/readers that it is not.

One aspect of this problem is that in my review I came up with several other tasks that have similar solutions. Another aspect is that other reviews appear to insist on a missing connection between lifelong VPR and the proposed approach, on which the authors do not appear to be able to provide a convincing response.

So maybe, instead of providing more baselines from other tasks or convincing arguments, the authors could just apply their approach to other tasks, as an efficient alternative to distillation. Even if CRLS is not significantly better, it is still a valid choice because it is efficient. Because distillation is ubiquitous, there is a abundance of such tasks. Two or three tasks of the authors' choice would do. If successful, the message would be that it is not very important if CRLS is really meant for lifelong VPR, because it benefits other tasks too.

I believe the paper would benefit a lot from the above suggestions if accepted. I am not sure if such updates are realistic for the camera-ready.

On other concerns:

- The authors did not address my original concern 2 (baselines for CSW).

- I am not sure what is the situation concerning other reviews about fair experimental settings - are all settings equal across all comparisons? It appears that the training set is still different.

**Questions:**

How much of the proposed line of work could the authors do? Could they present some additional concrete results and a plan for updating the paper? It is realistic to re-position the paper as suggested for ICLR?

---

> ### Author Response · Authors · 2024-11-27
>
> We would like to address the main concern you raised regarding the positioning of our work.
>
> We agree that there are several tasks, such as learning with noisy labels or semi-supervised learning, where the inconsistency between input examples and labels is a key challenge. In the context of visual place recognition (VPR), such inconsistency issues can primarily be arised due to viewpoint mismatches. While GPS information is generally accurate, if there is no viewpoint overlap between images captured at the same location, the visual cues can differ, leading to the inconsistency problem you mentioned. This ambiguity arises from viewpoint mismatches.
>
> In such cases, it is indeed appropriate to address the sample-specific ambiguity problem at the per-sample level. From this perspective, this issue has been studied in the VPR task, and the GCL [1] has tackled it by considering the viewpoint-induced ambiguity problem in VPR, solving it at the per-sample level.
>
> However, our work differs from these approaches in terms of both the motivation and the proposed approach. In lifelong VPR, the main challenge is NOT the inconsistency between input examples and labels during training, but rather the significant appearance changes of the same place over time during testing.
>
> Our proposed CRLS aims to learn a more robust representation that can accommodate these appearance changes by incorporating inter-class similarities. The goal is not to modify potentially incorrect labels during training, but to enable smoother transitions in the learned feature space, making the model more robust to the variations inherent in lifelong VPR.
> We appreciate your feedback and hope you will consider the clarification when evaluating our work.
>
> [1] Leyva-Vallina et al., Data-efficient Large Scale Place Recognition with Graded Similarity Supervision, CVPR 2023

---

> > ### Comment · Reviewer_MmfN · 2024-11-27
> >
> > I thank the authors for the response.
> >
> > I fail to follow the argument of training vs. testing. The proposed method uses a loss applied during training (based on modified labels) and a motivation and baseline is LS, also during training (also modified labels). What I suggested is more competitors at training.
> >
> > Learning something that is more robust at testing in whatever sense is still done by modifying the training process. What I suggested is looking for ideas at other tasks which modify the training process, even if their objective is different.

---

> > > ### Author Response · Authors · 2024-11-27
> > >
> > > We appreciate your suggestion and response.
> > >
> > > Label modification techniques (including label smoothing) have been extensively explored in diverse fields, such as calibration, generalization, un-/semi-supervised learning, and noisy label learning.
> > >
> > > We recognize that our approach is conceptually related to these areas. However, we would like to emphasize that **the VPR training dataset has been carefully curated to avoid inconsistencies (e.g., label noise)**, including considerations for viewpoint alignment, ensuring accurate GPS-tagged labels. Thus, existing studies in VPR have not aimed to use label modification techniques designed to address potential noise in given label information (although if un-/semi-supervised VPR were to emerge, such techniques could be applicable). Additionally, our method directly utilizes given class labels to perform class-adaptive label smoothing at the per-class level.
> > >
> > > In contrast to semi-supervised or noisy label learning, seminal approaches in those fields often:
> > >
> > > - Regress or classify the noise level (noisy or clean) of given labels (or pseudo-labels) [1, 2, 3, 4], which is not applicable in our case, as each sample has accurate GPS-tagged information.
> > > - Employ bootstrapping or distillation techniques, often requiring auxiliary networks or extensively modified architectures [4,5,6]. Such approaches would demand substantial changes to the standard VPR training pipeline, where a single network is typically trained on millions of images. In contrast, our method seamlessly integrates into this framework and demonstrates consistent improvements across various backbone networks, as validated by our experiments.
> > >
> > > We hope this clarification helps position our work appropriately in this context.
> > >
> > > [1] Li et al., DivideMix: Learning with Noisy Labels as Semi-supervised Learning, ICLR 2020
> > >
> > > [2] Kim et al., NLNL: Negative Learning for Noisy Labels, ICCV 2019
> > >
> > > [3] Li et al., Selective-Supervised Contrastive Learning with Noisy Labels, CVPR 2022
> > >
> > > [4] Han et al., Co-teaching: Robust Training of Deep Neural Networks with Extremely Noisy Labels, NIPS 2018
> > >
> > > [5] Ge et al., Mutual Mean-Teaching: Pseudo Label Refinery for Unsupervised Domain Adaptation on Person Re-identification, ICLR 2022
> > >
> > > [6] Cho et al., Part-based Pseudo Label Refinement for Unsupervised Person Re-identification, CVPR 2022

---

> > > > ### Comment · Reviewer_MmfN · 2024-11-28
> > > >
> > > > Dear authors,
> > > >
> > > > I am well aware that the problems are different. All I was saying was that solutions are related. Thus, you can try some baselines.
> > > >
> > > > Distillation is something very common. No new architecture is needed. It is common to use an exponential moving average of the student as a teacher, for example.
> > > >
> > > > I also mentioned methods using similarities between examples and prototypes, as well as pairwise similarities between examples.

---

> > > > > ### Author Response · Authors · 2024-12-01
> > > > >
> > > > > Dear Reviewer MmfN,
> > > > >
> > > > > We conducted additional experiments with Mean Teacher (MT) [A] and Neighbor Consistency Regularization (NCR) [B] as per the recommendation. While MT uses EMA-based distillation and NCR considers relationships between examples to prototypes and between examples, our CRLS approach focuses on the relationships between class prototypes themselves, not examples. The results validate that our CRLS approach outperforms these methods on VPR benchmarks as shown in Table 7 (in the main manuscript).
> > > > >
> > > > > | Method                    | SF-XL test v1 | SF-XL test v2 | MSLS Val | MSLS Chall. | Amster. |
> > > > > |---------------------------|---------------|---------------|----------|-------------|---------|
> > > > > | LR | 84.0          | 91.0          | 88.9     | 67.9        | 48.6    |
> > > > > | ACLS | 82.2          | 91.1          | 87.7     | 67.2        | 47.4    |
> > > > > | MT* | 84.4   | 90.5   |88.0  |   67.6  | 47.4  |
> > > > > | NCR* | 82.9   |90.6   |88.9  |    67.0 |48.7  |
> > > > > | **CRLS (Ours)**            | **86.0**      | **92.3**      | **90.1** | **68.8**    | **51.1**|
> > > > > `*` denote newly added methods.
> > > > >
> > > > > The effectiveness of our class-level approach can be attributed to the different characteristics of VPR datasets. With approximately 263k total classes, an average of 14k classes per training group, and around 20 instances per class, the scale and granularity of the class space in VPR training set is significantly different from typical datasets in other domains. Therefore, considering inter-class relationships may provide a sufficient and effective solution for the task.
> > > > >
> > > > > [A] Tarvainen et al., Mean teachers are better role models: Weight-averaged consistency targets improve semi-supervised deep learning results, NIPS 2017
> > > > >
> > > > > [B] Iscen et al., Learning with Neighbor Consistency for Noisy Labels, CVPR 2022

---

> > > > > > ### Comment · Reviewer_MmfN · 2024-12-01
> > > > > >
> > > > > > Dear authors,
> > > > > >
> > > > > > Thank you for the additional results. I believe this makes the narrative more solid.

---

> > > > > > > ### Author Response · Authors · 2024-12-02
> > > > > > >
> > > > > > > Dear Reviewer MmfN,
> > > > > > >
> > > > > > > We sincerely appreciate your confirmation that our additional results help strengthen the paper's narrative. We will reflect the discussed improvements in the main paper.
> > > > > > >
> > > > > > > Please let us know if you have any remaining concerns or suggestions.

---

### Official Review · Reviewer_sziE · 2024-11-03

**Soundness:** 3
**Presentation:** 2
**Contribution:** 2
**Rating:** 5
**Confidence:** 5

**Summary:**

This paper presents a Class-Relational Label Smoothing technology for the Visual Place Recognition (VPR) method with classification loss for training. The proposed Label Smoothing transforms one-hot hard labels into soft labels based on inter-class visual similarities and can bridge the task gap between classification and retrieval in VPR. To improve training stability, the authors also propose a Class Stability Weighting strategy, which dynamically adjusts the label smoothing process according to the stability of class weights.

**Strengths:**

1.	This paper is well-written and clearly presented.
2.	The proposed method is novel (in the VPR area) and well-motivated. The authors also provide derivative analysis for further explanation.
3.	Experiments are conducted on diverse benchmark datasets.

**Weaknesses:**

1.	Although the experimental results seem ok, the advantages compared to SOTA methods are not obvious. For example, on the commonly used Pitts30k/Pitts250k and MSLS val, this method doesn't outperform EigenPlaces with an obvious margin.
2.	The work follows the training setting of the EigenPlaces framework. However, this work uses a larger batch size and learning rate than EigenPlaces. The authors claim those do not enhance the performance of the baseline model. So why not use the exact same settings as EigenPlaces? Whether the benefits of the proposed method will become marginal under such a setting.
3.	Although the proposed method with the DINOv2 backbone seems to outperform other DINOv2-based methods (SelaVPR, SALAD, CricaVPR), their training datasets are different. The SF-XL dataset used in this work is the biggest one, so this comparison may be not fair. The results of this method seem to be worse than those of EffoVPR, but the latter uses a stronger backbone DINOv2-large. My suggestion is that the proposed method should be compared with EigenPlaces based on the DINOv2-base backbone.
4.	Minor error: In some places (Line 939, 940, 1019), EigenPlaces was mistakenly written as EigenPlace.
5.	All analysis in this paper is based on the CosFace loss function used in EigenPlaces/CosPlace. However, many recent VPR works are using multi-similarity loss in MixVPR. The contribution that this work can make to the VPR community seems limited.

**Questions:**

The results of two-stage methods (TransVPR, StructVPR, R^2Former, SelaVPR) seem much lower than the original papers. Did the authors only perform the first stage? This is unfair because the global features of these methods are more low-dimensional than others.

---

> ### Author Response · Authors · 2024-11-28
>
> >Weakness1.
>
> We conducted additional experiments using DINOv2 backbone on Pitts30k/250k. The results are included in Table 2 of the supplementary material.
>
> >Weakness2.
>
> We retrained our model using the same batch size and ResNet-50 as EigenPlaces. The performance remained similar, as shown below:
> | Method  | SF-XL test v1 | SF-XL test v2 | MSLS Val | MSLS Chall. | Amster. |
> |---------|---------------|---------------|----------|-------------|---------|
> | Ours    | **86.0**      | **92.3**      | 90.1     | 68.8        | 51.1    |
> | Ours*   | 85.2          | 91.9          | **90.3** | **69.6**    | **51.1**|
> Ours* denotes our model trained with EigenPlaces batch size.
>
>
> >Weakness3.
>
> To further examine the effectiveness of our method when applied to foundation models, we conducted additional experiments comparing our approach with naive label smoothing using the DINOv2 backbone. Table 4 in the supplementary material demonstrates that our method consistently outperforms naive label smoothing in this setting. These results suggest that our proposed CRLS can enhance the performance of foundation models in lifelong VPR tasks.
>
>
> >Weakness4.
>
> We have corrected the typo.
>
> >Weakness5.
>
> While metric learning loss such as multi-similarity loss is currently a popular choice for VPR tasks on moderately-sized datasets, our work focuses on classification-based losses due to their superior scalability. As the field progresses towards larger-scale datasets, the scalability advantages of classification-based methods will become increasingly important. In this context, our contribution using CRLS is particularly valuable, as it enables improved performance on large-scale VPR tasks.
>
> >Question.
>
> We reported the image retrieval results using only global descriptors without re-ranking (i.e., the first stage).
> Re-ranking technique is outside the scope of our current work, as our primary goal is to demonstrate the effectiveness of our proposed CRLS in learning better global representations for lifelong VPR. Nonetheless, we included the results of SelaVPR with re-ranking in Table 3 of the supplementary material.

---

> > ### Comment · Reviewer_sziE · 2024-12-01
> > **Official Comment by Reviewer sziE**
> >
> > Thanks for your response.
> > However, I still think that the proposed method has no obvious advantages over EigenPlaces. Table 2 of supplementary material only shows that the proposed work outperforms the previous DINOv2-based method (SALAD and BoQ). Since this paper uses a larger SF-XL dataset for training, it needs to be compared with DINOv2-based EigenPlaces to be convincing. Similarly, for the ablation study using the DINOv2-B backbone in Table 4 of supplementary material, why is there only naive label smoothing but not the EigenPlaces (DINOv2-based)? The recent work EffoVPR [1] is also based on the DINOv2 backbone and the EigenPlaces training framework, but its performance is better than this work.
> >
> > [1] Tzachor, Issar, et al. EffoVPR: Effective Foundation Model Utilization for Visual Place Recognition. 2024.

---

> > > ### Author Response · Authors · 2024-12-02
> > >
> > > Dear Reviewer sziE,
> > >
> > > As suggested, we have added EigenPlaces with DINOv2-B backbone to our ablation study. The updated Table 4 in the supplementary material now includes performance metrics with both naive label smoothing and EigenPlaces:
> > >
> > > | Method | Dim. | SF-XL test v1 | SF-XL test v2 | MSLS Val | MSLS Chall. | Amster. | Avg.
> > > |--------|------|---------------|---------------|----------|-------------|---------|---------|
> > > |EigenPlaces | 2048| 92.4| 93.0|91.1 |77.1 | 56.1| 81.9
> > > | LS     | 2048 | 91.1          | 93.3          | 91.2     | 76.5        | 59.1    | 82.2
> > > | Ours   | 2048 | **93.7**      | **94.0**      | **92.2** | **77.3**    | **59.9**| **83.4**
> > >
> > > Our approach shows a bigger margin of the average improvement, compared to the gain from LS to EigenPlaces.
> > >
> > > Regarding EffoVPR, which is a concurrent work, it has utilized a stronger backbone of DINOv2-large, as you mentioned. Nevertheless, our method appears to show competitive performance to their method.
> > > Furthermore, we believe our approach can be seamlessly integrated into the EffoVPR method by modifying its loss function, and
> > >  thus have an orthogonal contribution to this field.

---

### Official Review · Reviewer_fagY · 2024-11-03

**Soundness:** 2
**Presentation:** 4
**Contribution:** 3
**Rating:** 5
**Confidence:** 4

**Summary:**

The paper introduces a new model for visual place recognition (VPR) that incorporates a customized label-smoothing technique. By leveraging visual relationships between classes, the model is designed to learn a continuous representation space to manage various appearance changes, such as lifelong variations, that may often occur in VPR. In their label-smoothig approach the authors replace the traditional random class label smoothing with a similarity-based label smoothing. They note that label smoothing results in different class weights, where more challenging classes often require more training updates. To address this, they propose a method to identify these difficult classes and adjust their weights dynamically throughout training, ensuring training stability.

**Strengths:**

The paper tackles a specific aspect in VPR, that is lifelong changes that had not been explicitly addressed in prior research. The concept is effectively demonstrated, and the paper is well-written. The authors also analyze current benchmarks in this respect. The approach is novel and demonstrated over many experiments.

**Weaknesses:**

I have some concerns about the comparisons made in the paper and the rational behind the method.

1. The abstract's claim of "outperforming SoTA across 17 benchmarks" seems to be an overstatement. Current state-of-the-art (SoTA) methods typically rely on pretrained foundation models (e.g., SELA, SALAD, Crica, BoQ [1]). However, the paper includes only a limited number of experiments that compare against these leading techniques. While the paper does demonstrate the method's effectiveness across various architectures and benchmarks, it achieves SoTA results on only 5 benchmarks. To the best of my knowledge, on all the benchmarks used in the paper, SoTA is reached by foundation based models. I encourage the authors to extend their tests with foundation models to all the benchmarks used in the paper to better assess the effectiveness of label smoothing when applied to these SoTA techniques.

2.	Foundation models excel at capturing partial matches between images, potentially making label smoothing less impactful or redundant. When part of an image changes, current models, particularly those based on foundation models, are highly effective at identifying matches on the unchanged parts. This is why they also perform well in scenarios involving occlusion. It raises the question of what exactly CRLS captures in such cases—does it identify the changes and recognize what has been altered, or does it simply rely on the unchanged areas like other methods? It seems to me that the benefit of the suggested method is not really due to the moderate changes in the image, but rather due to smoothing of the classification space. I suggest the authors to provide attention maps that may shed light to this issue. I’d be glad to see examples of attention maps with foundation models against for instance SALAD, where the current model succeeds and SALAD fails.
3.	Another mechanism used to handle partial changes is re-ranking. It is unclear whether the results presented include re-rankers in the various methods compared. There are multiple approaches that incorporate re-ranking, such as SELA and R2Former. It is necessary IMO, to see results with re-ranking of those methods (and others) and also on benchmarks with occluded benchmarks. Table 4, is particualry important as it includes datasets with more significant changes, such as day vs. nights, weather changes etc. I’d suggest to compare the suggested method against foundation models on these datasets. I believe that most of the results can be found in the literature.
4.	Regarding the reported results, the model performs worse on MSLS - R@5 when using foundation models (as shown in Table 11 - Appdx). It shows better performance on SF, which it was trained on. In the single-view category, the results are significantly behind MixVPR, which is attributed to the use of longer features. However, since the authors add extra layers to foundation-based models, I’d suggest to train their model with additional layer to obtain the same feature size, for comparison. It is not clear if the observed differences are solely due to feature size.

[1] Ali-bey etal, BoQ: A Place is Worth a Bag of Learnable Queries, CVPR 2024

Comments:

$l$ is EQ. 1 is not defined. It will be fair to state in the comparison with AnyLoc that it is a zero-shot approach (without training on VPR data).

**Questions:**

1.	How will it work on Nordland, with seasonal changes?
2.	It is not clear what is the training dataset? Which data has been used?
3.	For DINOv2 based foundation model, the authors apply two-layer MLPs for token embeddings, followed by a fully connected layer for the dimensionality reduction. Is there any ablation study showing that the performance boost is not due to these architectural changes?

---

> ### Author Response · Authors · 2024-11-27
>
> Thank you for your valuable feedback and comments. We would like to address your concerns raised in the weaknesses and provide additional information.
> 1. We have conducted experiments using foundation models on all benchmarks, and the results including BoQ-DINOv2 are provided in the supplementary material (see Table 1-3).  We also confirmed that our method using the foundation model obtains superior results in other benchmarks.
>
> 2. Our method aims to capture the continuity of global similarity rather than focusing on specific local regions. Therefore, we exhibit t-SNE visualization (Figure 1 of the supplementary material) to demonstrate the continuous representation space learned by CRLS, instead of using attention maps.
>
> 3. We have added the re-ranking accuracy for SELA on the single-view datasets in Table 3 of the supplementary material.
>
> 4. In addition to the 2048-dimensional features used in our method, we have included experiments with 4096 and 8192 dimensions using the foundation model (see Tables 1-3 of the supplementary material).
>
> Regarding the comments, we revise the paper highlighting changes in blue.
>
> To answer your questions:
>
> >Q1.
>
>  The Nordland dataset primarily represents seasonal changes, which are actual changes similar to lifelong variations. Our method also appears to handle these changes effectively, evidenced by the results of Nordland in Table 3 of the supplementary material .
>
> >Q2.
>
> As mentioned in L374 in the main manuscript, we utilized the SF-XL dataset for training our models.
>
> >Q3.
>
> We have added Table 4 in the supplementary material that compares the performance of a foundation model trained with naive label smoothing and our method (CRLS). The results show that CRLS consistently outperforms naive label smoothing.

---

> ### Comment · Reviewer_fagY · 2024-11-28
>
> I would like to thank the authors for their response and additional results. I still have some concerns.
>
> Q1: I must admit that I find it difficult to understand the point you are trying to convey with the t-SNE figure in SM. The result in your approach (c) appears to be the same as (a) with hard labels. My interpretation is that you aim to demonstrate that images undergoing gradual changes (possibly over time) are progressively moving farther from their original state. However, I’m afraid this is not reflected in the figure.
>
> Q2: This leads me to the question for definition of lifelong change. Are we expecting a gradual transformation, similar to aging (in human)? I believe temporal changes typically result from factors like construction in front of the house, or the addition of a new building on the side (which doesn’t correspond to the original scene image), as can be seen in Fig. 1. This of course can also stem from variations in viewpoint. How, then, do we define what constitutes a gradual departure from the original scene appearance?
>
> Q3: Regarding Table 4 in SM, it mentions R1 and R5, but there seems to be only a single result presented there. Could you clarify?

---

> ### Author Response · Authors · 2024-11-29
>
> We would like to thank the reviewer for the additional feedback, and address the question as follows.
>
> >Q2.
>
> Lifelong variations involve **actual** changes or modifications to a place over time, unlike other variations. Rather than explicitly modeling lifelong changes of individual places during training, our method learns a continuous representation space that captures similarities between different, but visually related, places. These similar yet distinct places can effectively simulate the types of actual changes a single place might face in lifelong scenarios. This allows the model to implicitly handle significant appearance changes without requiring a precise definition or examples of gradual transformations for each place in the training data.
>
>
> >Q1.
>
> Our t-SNE visualizations are intended to illustrate the continuous representation space learned by CRLS, focusing on the distinctness and grouping of visually similar classes. The t-SNE algorithm emphasizes this distinctness rather than showing gradual changes.
>
> In (a) hard label, buildings that share highly similar visual characteristics are separated into totally different clusters, despite their visual similarities. In contrast, (c) CRLS maintains more unified clusters for these visually similar buildings while preserving appropriate separation between different places. This clustering of similar places enables our model to be more robust to the actual changes that occur in lifelong scenarios.
>
> >Q3.
>
> We apologize for the confusion in Table 4. All additional experimental results provided for discussion, including those in Table 4, report Recall@1 for consistency.

---

> ### Author Response · Authors · 2024-12-03
>
> Dear Reviewer fagY,
>
> It appears that you have posted your comment in the thread for Reviewer EqJa.
> We have addressed the questions you raised, but we're wondering if you have any additional concerns regarding our responses.
>
> To provide further clarification, we believe that leveraging similar but different places to simulate actual modifications is fundamentally different from viewpoint variations. In the case of viewpoint variations, the actual object itself remains unchanged.

---

### Official Review · Reviewer_EqJa · 2024-11-04

**Soundness:** 3
**Presentation:** 3
**Contribution:** 3
**Rating:** 6
**Confidence:** 3

**Summary:**

The paper addresses the problem of visual place recognition, where the goal is given a query image and a database of images at different locations to localize the query by matching it to the database.  Building on previous work, this task is formulated as a classification task where the individual places are classes, and the goal is to classify the query image into one of the possible classes (places). The first contribution of this work is to use a label smoothing regularization strategy that assigns non-zero probability during training to  visual places other than the ground truth places with larger probability assigned to places that are visually similar. The second contribution is to adjust the weight of the label smoothing regularization weight for each individual class during training. Results are shown on several place recognition benchmark and demonstrate improvements over prior methods.

**Strengths:**

- Novelty: using the inter-class relations to adjust the label smoothing is a nice idea, which makes sense for place recognition and seems novel in the place recognition context.

- Results: the proposed approach demonstrates relatively small but consistent improvements over prior methods in many benchmarks.

**Weaknesses:**

- 1. The clarify of the second contribution. (L268): The paper describes “min-max” normalization of the magnitudes over all classes”. However, it is not clear what is exactly meant here. Please describe exactly the step-by-step procedure that is used for this normalization. Please include a motivation for each step. Please link this procedure to the analysis shown in the equation 9 and in the supplementary material. This is important to justify/clarify the second contribution of the paper.

- 2. L250: “We assign labels in a sequence that mirrors the visual similarity to the target class, thereby enhancing the model’s ability to generalize across visually similar scenarios, e.g., lifelong variation.” The idea of using the intra-class visual similarity for label smoothing makes sense and is well explained and justified. But it is not clear how it relates to lifelong variation. Why should it improve generalization across lifelong variations? Please clarify this point.

**Questions:**

- Please address the questions 1 and 2 above.

- 3. [Optional] It could be interesting to do error bars for the presented results by bootstrapping, which is a standard approach in other application areas of machine learning such as bioinformatics, see e.g. : https://www.dummies.com/article/academics-the-arts/science/biology/the-bootstrap-method-for-standard-errors-and-confidence-intervals-164614/

 I understand that this is not standard in the place recognition research and for the place recognition benchmarks (hence it is marked as optional here), but it would be interesting to start reporting those when comparing the different methods or ablations going forward.

---

> ### Author Response · Authors · 2024-11-27
>
> >Q1.
>
> As shown in Figure 3, classes with lower magnitudes tend to be easier to learn and are more likely to be well-trained. The key contribution of CSW lies in its ability to focus on these low-magnitude classes, which mitigates potential instabilities of training with CRLS. Furthermore, while low-magnitude classes may not significantly contribute to the learning process due to their already well-learned state, CRLS encourages these classes to participate in the learning by considering inter-class visual similarities. This allows for a more comprehensive utilization of all classes during training.
> Regarding min-max normalization, we chose this approach for its simplicity in converting magnitudes into weights. Specifically, we compute the loss weight $\gamma_{y_i}$​, bounded between 0 and 1, is computed as follows:
> $$\gamma_{y_i} = \frac{||W_{y_i}|| - \min_j ||W_j||}{\max_j ||W_j|| - \min_j ||W_j||}$$
>
> >Q2.
>
> There is some overlap in the general response. In brief, CRLS improves generalization across lifelong variations by learning a continuous representation space that captures the visual similarities between classes. This allows the model to handle significant appearance changes more effectively compared to methods that rely solely on discriminative features.
>
> >Q3.
>
> We understand your intention to control for randomness in the results by suggesting bootstrapping for error bars. However, given the extensive experiments already conducted and the large scale of the datasets used, bootstrapping for all results would be quite challenging. Despite this, we find your suggestion intriguing and will consider incorporating it in future research to provide a more comprehensive analysis of the results.

---

> > ### Comment · Reviewer_EqJa · 2024-11-30
> > **Partial clarification**
> >
> > Dear Authors,
> >
> > Thank you for the response.
> >
> > I have read both the common response and the response to my review. I am still not very clear about the motivation and the intuition of the proposed approach. Also, I still do not quite understand the reasoning behind the provided loss weight gamma and why it should be beneficial in the loss function.
> >
> > Thank you.

---

> > > ### Author Response · Authors · 2024-12-02
> > >
> > > Dear Reviewer EqJa,
> > >
> > > We would like to provide a concise summary of the key motivations and intuitions behind the two main components of our approach.
> > >
> > > CRLS (Class-Relational Label Smoothing):
> > > - Enables learning representations that are robust to the significant appearance changes places undergo over time in lifelong VPR scenarios
> > > - Incorporates inter-class visual similarities during training to encourage the learned feature space to be continuous
> > > - Maps visually similar places to nearby points in the feature space, even if they are distinct locations, while still maintaining discriminative clusters for each place
> > > - Utilizing the similar yet distinct places can effectively simulate the types of actual changes a single place might face in lifelong scenarios
> > >
> > > CSW (Class Stability Weighting):
> > > - Addresses potential instability in the training dynamics caused by using the evolving class weights to construct the soft labels in CRLS
> > > - Dynamically adjusts the influence of CRLS for each class based on the class's stability, which is quantified by the magnitude of its weight vector
> > > - Emphasize more on CRLS to classes that are more stable and well-learned, as indicated by their smaller weight magnitudes
> > > - Emphasize less on CRLS to classes that are instable and need to learn more discriminative features due to lower accuracy (Fig 3 in the main manuscript), as indicated by their higher weight magnitudes
> > > - Balances discriminative power and regularizing effect of CRLS
> > >
> > > If any part of the summarized explanation requires further clarification or additional details, please let us know.

---

> > > > ### Comment · Reviewer_fagY · 2024-12-03
> > > > **Posing the paper as lifelong task**
> > > >
> > > > The problem I believe is the way the paper is posed. The proposed solution does not directly address the lifelong learning challenge, as the changes you describe could occur also from a different viewpoint. Your training scheme enforces smoothness in the embedding space, which is also evident from the t-SNE visualization provided in the supplementary material. Ultimately, the clusters represent visually similar locations rather than the same place across different timestamps.

---

> ### Author Response · Authors · 2024-12-03
>
> Dear Reviewer fagY,
>
> We acknowledge that the visual changes can occur due to both viewpoint variations and lifelong variations. However, lifelong variations are more challenging to address as they involve actual modifications, which differ from viewpoint variations.
>
> For addressing viewpoint variations, we explicitly incorporate them into the training data to ensure robustness [A]. With Google Street View, exact viewpoints can be defined, and the expected changes in images due to viewpoint variations are relatively predictable.
>
> In contrast, for lifelong variations, it is challenging to explicitly include them in the dataset because not all buildings are guaranteed to change over time. Therefore, we indirectly simulate lifelong variations by leveraging the relationships between similar places. These similar places can effectively simulate actual changes that may occur over time, unlike viewpoint variations. This approach aims to improve the generalization performance for actual modifications such as lifelong variations.
>
> The challenging nature of real-world lifelong variations can be observed in Figure 7 of the main manuscript, which shows examples from the AmsterTime and SF-XL test v1 datasets.
>
> [A] Berton et al., EigenPlaces: Training Viewpoint Robust Models for Visual Place Recognition, ICCV 2023

---

### Author Response · Authors · 2024-11-27
**Clarification of the proposed method**

We uploaded **supplementary material** for additional results.

Regarding the effectiveness of CRLS in addressing lifelong variations, we would like to provide further clarification.

Existing methods for visual place recognition predominantly focus on learning highly discriminative features, either through enforcing hard decision boundaries between places in classification-based approaches or by maximizing inter-class distances in the feature space through metric learning losses. While these techniques excel at distinguishing between different places under static conditions, they can struggle to handle the significant appearance changes that occur in lifelong scenarios. The emphasis on learning over-discriminative features can lead to overfitting to specific visual appearances, causing models to incorrectly treat modified places as entirely different locations in lifelong scenarios where appearance or structure can drastically change.

To address this limitation by alleviating over-discrimination and promoting continuous representation space, we propose CRLS, a method that leverages the visual similarities between classes during training. By incorporating inter-class relationships, CRLS allows the model to preserve smooth visual transitions while maintaining discriminative power.

As shown in **Figure 1 of the supplementary material**, our CRLS learns a visually continuous feature space where similar-looking places form smooth transitions rather than discrete clusters. This continuity enables our model to effectively handle severe appearance changes such as building renovations. For instance, class cluster 3 shows different buildings that share similar structural characteristics - while these are different places rather than actual renovations from the same building, our method to establish continuous relationships between such visually similar buildings can simulate how real building renovations should be handled in lifelong scenarios.

Also, it is important to note that CSW helps to balance the discriminative power and visual continuity in the representation space. By assigning higher weights to low-magnitude classes, which are already discriminative, CSW allows CRLS to focus more on capturing visual continuity. This balance is crucial for accurate place recognition while handling the challenges posed by lifelong variations.

We hope this clarification has helped your evaluation of our work.

---

### Note · Authors · 2025-02-04

I have read and agree with the venue's withdrawal policy on behalf of myself and my co-authors.